# Genetic Biomarkers Associated with Dynamic Transitions of Human Papillomavirus (HPV) Infection–Precancerous–Cancer of Cervix for Navigating Precision Prevention

**DOI:** 10.3390/ijms26136016

**Published:** 2025-06-23

**Authors:** Pallop Siewchaisakul, Jean Ching-Yuan Fann, Meng-Kan Chen, Chen-Yang Hsu

**Affiliations:** 1Faculty of Public Health, Chiang Mai University, Chiang Mai 50200, Thailand; pallop.s@cmu.ac.th; 2Department of Health Services Administration, College of Public Health, China Medical University, Taichung 406040, Taiwan; jeanfann@mail.cmu.edu.tw; 3Department of Family Medicine, National Taiwan University Hospital Hsinchu Branch, Hsinchu 302058, Taiwan; 4Master of Public Health Program, College of Public Health, National Taiwan University, Taipei 100025, Taiwan; 5Taiwan Association of Medical Screening, Taipei 103015, Taiwan

**Keywords:** cervical cancer, HPV, genetic factors, epigenetic factors, multistate disease evolution, nature history

## Abstract

Precision prevention strategies for cervical cancer that integrate genetic biomarkers provide opportunities for personalized risk assessment and optimized preventive measures. An HPV infection–Precancerous–Cancer risk assessment model incorporating genetic polymorphisms and DNA methylation was developed to better understand the regression and progression of cervical lesions by HPV infection status. Utilizing a virtual cohort of 300,000 Taiwanese women aged 30 years and older, our model simulated the natural history of cervical cancer, capturing transitions from a healthy state through precancerous lesions (LSILs and HSILs) to invasive carcinoma and incorporating the possibility of regression between states. Genetic and epigenetic markers significantly influenced disease transitions, demonstrating heterogeneous risks among women with distinct molecular biomarker profiles. Guided by these individual risk profiles, tailored preventive strategies including varying intervals for Pap smear screening, HPV DNA testing, and HPV vaccination showed improved efficiency and effectiveness in reducing cervical cancer incidence compared to uniform approaches. The proposed dynamic transition model of cervical neoplasms incorporating genetic biomarkers can facilitate the development of an individualized risk-based approach for guiding precision prevention towards the goal of cervical cancer elimination.

## 1. Introduction

Since the 1960s, population-based cervical cancer screening using the Papanicolaou (Pap) smear has significantly reduced cervical cancer mortality rates and has been adopted as a mainstay of prevention worldwide [1,2]. Following the identification of human papillomavirus (HPV) as a primary oncogenic virus in the 1980s, HPV testing and vaccination were subsequently incorporated into cervical cancer prevention strategies [3,4,5,6,7,8,9]. Recent studies have further elucidated that the integration of HPV genes into cervical epithelial cells in women with persistent infections disrupts protein expression and activates carcinogenic pathways [10,11,12]. Concurrent advancements in the understanding of genetic and epigenetic factors associated with cervical cancer risk have underscored the importance of monitoring low-grade squamous intraepithelial lesions (LSILs), high-grade squamous intraepithelial lesions (HSILs), and persistent HPV infections [10].

These insights into the evolution of cervical cancer support the development of a more comprehensive prevention framework that combines HPV vaccination, HPV testing, and the established role of Pap smears to achieve the goal of cervical cancer elimination by 2030 [13,14]. Furthermore, the multifactorial nature of cervical carcinogenesis, including virological factors, genetic susceptibility, and individual characteristics, presents an opportunity to develop personalized prevention strategies. While current prevention strategies primarily focus on risk assessment based on HPV infection status [15], the integration of individual genetic and epigenetic characteristics to refine risk-guided approaches remains limited. Developing an HPV infection–Precancerous–Cancer risk assessment model that incorporates genetic and epigenetic markers is essential for advancing the full spectrum of interventions, including vaccination, HPV testing, and Pap smears, and for enhancing cervical cancer prevention through tailored, risk-based approaches.

This study aims to develop a mathematical model that integrates HPV infection status, genetic alterations, and epigenetic biomarkers to simulate the natural history of cervical neoplasia. The model provides a foundation for designing precision prevention strategies that combine Pap smear screening, HPV DNA testing, and HPV vaccination. To elaborate, this study was conducted in three steps:Characterizing Risk Determinants: We first examined the role of HPV infection status along with genetic and epigenetic markers in influencing the progression from LSILs and HSILs to invasive cervical cancer.Modeling Disease Evolution: These molecular factors were then embedded into a virtual cohort, representative of Taiwanese women, to simulate lesion progression using published natural history parameters [2,3].Evaluating Prevention Strategies: Finally, the effectiveness of different combinations of prevention modalities—HPV vaccination, HPV testing, and Pap smears at varied intervals—was assessed across distinct molecular risk profiles.

The relationship between genetic/epigenetic changes and prevention strategies is mediated through their impact on individual risk percentiles. Women with high-risk molecular profiles benefit more from intensive prevention (e.g., combined HPV testing, vaccination, and annual Pap smear), while those at low risk may safely extend screening intervals.

## 2. Results

### 2.1. Natural Evolution of Cervical Cancer

#### 2.1.1. The Natural History Model of Cervical HPV Infection-Precancerous-Cancer Process

We developed a seven-state natural history model for delineating the HPV infection–Precancerous–Cancer process as illustrated in Figure 1. Starting from the healthy state (healthy women), individuals may progress through LSILs (CIN1), HSILs (CIN2/3), and further to invasive cervical carcinoma (upper part of Figure 1). Given the well-established role of HPV infection in cervical cancer progression [4,13,14], the constructed natural history model allows for the transition from a healthy state to HPV-infected status, which further accelerates the progression of cervical cancer from LSILs and HSILs to invasive carcinoma (lower part of Figure 1).

Note that the proposed seven-state natural history model also allows for regression between states prior to evolving into invasive carcinoma. For example, regression from HSILs to LSILs and from LSILs to a healthy state is possible in the model. Similarly, regression from an HPV-infected state to a healthy woman’s state is also considered.

#### 2.1.2. Genetic-Biomarker-Driven Cervical Cancer Evolution

Beyond HPV infection status, cervical cancer progression is influenced by multiple molecular factors, including genetic polymorphisms and epigenetic modifications. As illustrated in Figure 1, polymorphism genes such as CD28 (rs3116496), IFNG (rs2430561), Pre-miR (rs11134527), and LAMB3 (rs2566), along with DNA-repairing SNPs, affect individual susceptibility to cervical cancer [16,17,18,19]. Table 1 provides a detailed list of genetic markers and their respective effect sizes on disease progression, from a healthy state or an HPV-infected state to LSILs and beyond. The population proportion of each genetic marker is also listed in Table 1. While certain SNPs contribute to disease progression, others are associated with persistent HPV infection [20]. In addition to SNPs, DNA methylation markers such as CCNA1, C13ORF18, and SFRP play significant roles in regulating the progression of cervical neoplasms, with promoter methylation also associated with cervical carcinogenesis [21,22,23,24]. These genetic signatures influence multiple transition steps in the evolution of cervical cancer. Table 2 summarizes the effect sizes of state-specific genetic influences on cervical cancer progression, along with the population proportions of relevant genetic markers.

Building on the molecular data presented above—including the population distribution and state-specific effect sizes of each genetic and epigenetic biomarker—we constructed a simulated cohort of 300,000 women, embedding these molecular characteristics within the natural disease course of cervical neoplasia. This virtual cohort reflects the progression from a normal epithelium to an LSIL, HSIL, and ultimately invasive cervical carcinoma. To evaluate the influence of HPV infection status and molecular risk factors on cervical lesion progression, we integrated data abstracted from the literature into this representative cohort of Taiwanese women. Utilizing the natural history parameters reported by Chen et al. (2011) [3], we projected the disease burden in the absence of preventive interventions, thereby establishing a baseline scenario for the subsequent evaluation of precision prevention strategies.

### 2.2. Cervical Cancer Risk Projected by Genetic Biomarker-Guided Seven-State Natural History Model

#### 2.2.1. Internal and External Validation

As shown in Appendix A, the model yielded 2364 expected cervical cancer cases in the simulated Taiwanese cohort, closely matching the 2386 cases derived from empirical inputs. The resulting goodness-of-fit test yielded χ^2^ = 0.21 (*p* = 0.65), indicating an adequate internal model fit.

For external validation using the Japanese female cohort scenario, the model estimated 1793 cervical cancer cases, compared to the empirical estimate of 1787 cases, yielding a χ^2^ = 0.02 (*p* = 0.88). This result supports the generalizability and external validity of our dynamic model across populations with differing demographic and screening profiles.

These validation findings affirm the robustness of our model under both simulated and independent real-world conditions and support its applicability for guiding precision prevention strategies in diverse settings.

#### 2.2.2. Population Risk Stratification

Supported by the established multistate and multifactorial model, the cervical cancer risks were estimated across a spectrum of percentiles of risk scores for the simulated population. The effect of population risk stratification on cervical cancer is presented in Figure 2, with the relative risk (RR) shown in Figure 2a (compared with women at the median risk score, or 50th percentile) and lifetime risk estimates in Figure 2b. The lifetime risk estimates for cervical cancer varied considerably across different risk groups, with RR ranging from 0.32 to 9.53 (Figure 2a) and lifetime risk ranging from 0.11% to 2.49% (Figure 2b).

At the 90th percentile of the risk score distribution, women had an RR of 9.53 (Figure 2a) and a lifetime cervical cancer risk of 2.49% (Figure 2b). In contrast, those in the 10th percentile of score had a lower relative risk of 0.32 (Figure 2a) and a lifetime risk of 0.11% (Figure 2b). This marked gradient in cervical cancer risk highlights the heterogeneity of the population and emphasizes the value of risk stratification by using the developed multistate and multifactorial score in guiding personalized cervical cancer prevention strategies. Detailed results on RR and lifetime risk are provided in Appendix A.

#### 2.2.3. Scenarios of Personalized Risk Assessment

Using the population distribution and effect sizes of relevant genetic and epigenetic markers, we generated a virtual cohort of 300,000 women that simulated the natural progression from normal cervical epithelium through the LSIL and HSIL to invasive cervical carcinoma. To assess the influence of HPV infection status, genetic polymorphisms, and DNA methylation markers on the progression of cervical neoplasia, we incorporated molecular data from the literature into the model. This virtual cohort was constructed to be representative of the Taiwanese population. Natural history parameters reported by Chen et al. (2011) [3] were used to estimate disease progression in the absence of preventive interventions, incorporating individual-level molecular profiles.

To illustrate this dynamic transition from HPV infection to precancerous lesions and finally to cervical cancer, six representative cases were selected (Figure 3, Cases A to F). The first three cases (Cases A, B, and C; Figure 3a) represent women with HPV infection, while the remaining three (Cases D, E, and F; Figure 3b) are HPV-negative. The complete genetic and DNA methylation profiles of these six individuals are summarized in the profile in Table 3.

##### Women with HPV Infection

Here we show the individual evolution of cervical cancer risk using six cases, three with HPV infection (Figure 3a, Case A to Case C) and three without HPV infection (Figure 3b, Case D to Case F). These cases illustrate the impact of the combined effects of SNP genetic biomarkers and DNA methylation, along with HPV infection status, on cervical cancer development. For a woman with HPV infection, the SNP genetic biomarker of Pre-miR-218 expression and C13ORF18/DAPK DNA methylation (Case A, orange line in Figure 3a) had a remarkable higher probability of having cervical cancer compared with the women with HPV infection, the SNP genetic biomarker of LAMB3 expression only (Case C, green line in Figure 3a). The cervical cancer risk for Case B (HPV infection, SNP genetic biomarker of Pre-miR-218 and LAMP3 expression, and DNA methylation of DAPK, HIC-1, and RAR-beta (Yellow line in Figure 3a)) is close to that of Case A.

##### Women Without HPV Infection

For a woman without HPV infection (Figure 3b), the probability of developing cervical cancer is remarkably lower compared to women with HPV infection. However, combinations of SNP genetic markers and DNA methylations still contribute to the variations in the probability of developing cervical cancer risk.

The detailed molecular profiles for the six illustrative cases are listed in Table 3 alone with the raw risk scores and the percentiles of risk level. To facilitate interpretation, we also provide distributions of the overall risk score, as well as stratified distributions for HPV-positive and HPV-negative individuals, in Appendix A.

### 2.3. Precision Cervical Cancer Prevention Strategies

Based on the individual risk percentiles derived from molecular profiles, the relative effectiveness of various cervical cancer prevention strategies was evaluated across six illustrative cases. Cases A and B, both exceeding the 80th risk percentile, were classified as high-risk. For these cases, the combination of HPV vaccination, HPV testing, and annual Pap smears resulted in a significant risk reduction of 39% compared to the standard triennial Pap smear strategy (relative risk [RR]: 0.61; 95% confidence interval [CI]: 0.53–0.71; *p* < 0.001).

For women at intermediate risk levels—represented by Case C (40–60th percentile), Case D, and Case E (both 60–80th percentile)—alternative prevention strategies combining either HPV testing with triennial Pap smear or HPV vaccination with triennial Pap smear also demonstrated statistically significant risk reductions. Detailed relative risks and confidence intervals for these strategies are reported in Table 3 of the revised manuscript.

In contrast, for low-risk individuals such as Case F (<20th percentile), extending the inter-screening interval to six years yielded a comparable level of protection to the standard triennial Pap smear (RR: 1.02; 95% CI: 0.71–1.46; *p* = 0.93), indicating no significant increase in risk.

These findings support the feasibility of tailoring cervical cancer prevention strategies based on molecular risk stratification. Guided by the risk profile of each woman captured by the percentile of risk score distribution, a personalized cervical cancer prevention strategy can be implemented. Table 4 shows the effectiveness of cervical cancer prevention strategies in terms of RR covering the combinations of HPV testing, Pap smear, and HPV vaccination for women across the spectrum of risk strata.

#### 2.3.1. Low-Risk Group

Women with a risk score percentile of less than 20%, whose lifetime cervical cancer risk ranges between 0.11% and 0.22%, were categorized into the low-risk group. In the absence of HPV testing, annual, triennial, and 5-yearly Pap smear screening resulted in RRs of 0.37 (95% CI: 0.28–0.40), 0.45 (95% CI: 0.34–0.58), and 0.44 (95% CI: 0.34–0.57), respectively, compared with a baseline RR of 1.00 for the scenario without any prevention strategy.

Incorporating a one-shot HPV test further reduced the RRs to 0.31 (95% CI: 0.28–0.39), 0.35 (95% CI: 0.30–0.41), and 0.42 (95% CI: 0.37–0.49), respectively. The addition of HPV vaccination to Pap smear screening led to a lower RRs of 0.32 (95% CI: 0.23–0.46), 0.32 (95% CI: 0.23–0.46), and 0.39 (95% CI: 0.28–0.54) for annual, triennial, and 5-year screening, respectively. When combined with HPV testing, the RRs were further reduced to 0.28 (95% CI: 0.20–0.40), 0.29 (95% CI: 0.20–0.41), and 0.34 (95% CI: 0.25–0.48), respectively.

In contrast, HPV vaccination alone conferred a modest RR of approximately 0.94% CI: 0.73–1.20) and 0.96 (95% CI: 0.76–1.21) for the scenarios without and with HPV testing, respectively. The marginal effect of HPV vaccination conferred by HPV vaccination was estimated at 3% to 8% for the low-risk women.

#### 2.3.2. Intermediate Risk Group

Women with risk scores between the 20th and 80th percentiles, corresponding to lifetime cervical cancer risks ranging from 0.22% to 1.69%, were categorized into the intermediate-risk group. In this group, annual, triennial, and five-yearly Pap smear screenings resulted in RRs of 0.33–0.44, 0.35–0.46, and 0.43–0.50, respectively. The addition of HPV vaccination to Pap smear screening further reduced these RRs to 0.15–0.24 (annual), 0.24–0.26 (triennial), and 0.25–0.28 (five-yearly).

Incorporating one-time HPV testing into these combined strategies modestly reduced the RRs across most scenarios. When HPV vaccination was implemented as the sole preventive measure, the RRs achieved by prevention strategies were moderately reduced, ranging from 0.28 to 0.50 with Pap smear screening alone, and from 0.19 to 0.26 when Pap smear screening was combined with HPV vaccination. The marginal effect of vaccination was estimated to range from 8% to 22%, showing an increasing marginal benefit among women at higher cervical cancer risk.

#### 2.3.3. High-Risk Group

Women with risk score percentiles above the 80th percentile, corresponding to a lifetime cervical cancer risk ranging from 1.69% (80th percentile) to 2.49% (90th percentile), were categorized into the high-risk group. The RRs of cervical cancer for women in this high-risk group receiving HPV vaccination alone were estimated at 0.73 (95% CI: 0.66–0.81) without HPV DNA testing and 0.70 (95% CI: 0.63–0.77) with HPV DNA testing. When combined with triennial Pap smear screening, the estimated RRs further decreased to 0.36 (95% CI: 0.32–0.41) without HPV DNA testing and 0.32 (95% CI: 0.28–0.36) with HPV DNA testing. In comparison, the use of Pap smear screening alone (without HPV vaccination) in the high-risk group resulted in RRs of 0.42–0.54 (without HPV DNA testing) and 0.44–0.48 (with HPV DNA testing), depending on the screening interval. The marginal effect of vaccination was estimated to be approximately 13–18%.

## 3. Discussion

### 3.1. Genetic-Biomarkers-Supported Cervical HPV Infection-Precancerous-Cancer Process

We developed a novel mathematical model integrating multistate progression and the regression of cervical cancer states, ranging from healthy condition through precancerous lesions (CIN1, CIN2/3) to invasive cervical carcinoma, explicitly incorporating HPV infection dynamics. The proposed model advances prior work by simultaneously accounting for heterogeneous genetic and epigenetic biomarkers influencing disease susceptibility and progression [25]. Note that the inclusion of regression pathways reflects clinical observations, as cervical lesions, especially in their early stages, may regress spontaneously, highlighting the nonlinear and potentially reversible nature of cervical carcinogenesis.

Multiple approaches have been proposed to advance the goal of precision and personalized cervical cancer prevention, particularly through the integration of artificial intelligence and predictive modeling. For instance, Wang et al. [26] developed the AI-based Cervical Cancer Screening System (AICCS), which applied deep learning to enhance diagnostic accuracy in cytology grading, demonstrating improved sensitivity over conventional Pap smear interpretation. He et al. [27] employed machine learning to predict the spontaneous regression of LSILs, highlighting non-traditional risk factors such as sleep quality, thus expanding the scope of individualized risk assessment. Rothberg et al. [28] proposed a demographic-based risk prediction model that utilized routine clinical variables to personalize cervical screening intervals by predicting CIN2+ risk. Elvatun et al. [29] conducted a comparative evaluation of existing cervical cancer risk prediction models across diverse populations, revealing heterogeneity in model performance but identifying several algorithms with promising cross-population generalizability. Additionally, Dong et al. [30] introduced the SMART-HPV model, which integrated high-risk HPV (hrHPV) genotyping to stratify risk with high accuracy, particularly in settings with limited screening access.

While these models represent significant progress, most focus on binary classification tasks—predicting the presence or absence of specific endpoints such as CIN2+ or cervical cancer—limiting their applicability in simulating lesion evolution and in guiding early-stage preventive interventions.

By contrast, the current study introduces a dynamic, multistate natural history model of cervical neoplasia progression, spanning from normal epithelium through LSILs and HSILs to invasive carcinoma. Our model uniquely incorporates genetic and epigenetic biomarker profiles, enabling risk stratification beyond demographic or HPV genotype alone. This structure supports the simulation of personalized prevention strategies—including HPV vaccination, testing, and Pap smear screening—tailored to individual molecular risk percentiles.

The key innovation lies in the model’s ability to simulate temporal disease progression under various intervention scenarios, allowing for the evaluation of both early prevention and screening optimization strategies. This approach addresses a critical gap in existing models by offering a more granular and biologically informed framework for precision cervical cancer prevention, thus providing actionable insights for both clinical decision making and public health planning.

### 3.2. Molecular Markers on the Risk of Cervical Cancer

Genetic susceptibility to cervical low-grade squamous intraepithelial lesions (LSILs) and their subsequent progression to high-grade lesions (HSILs) and invasive cervical cancer involves complex immunological, molecular, and cellular mechanisms. Notably, immunoregulatory genes located on chromosome 2q33 have been identified as significant contributors to cervical cancer susceptibility. Specific polymorphisms, such as CD28+17 (TT) and IFNG+874 (AA), have been associated with an increased risk of LSILs, underscoring the role of host immune modulation in early lesion development [18]. MicroRNA-related pathways also contribute to cervical carcinogenesis. The interaction between miR-218 and LAMB3 has been implicated in lesion progression. Specifically, the HPV-16 E6 oncoprotein downregulates miR-218, leading to the overexpression of LAMB3, which in turn facilitates HPV persistence and promotes the development of precancerous lesions such as LSILs [16]. Furthermore, genetic variants in apoptosis-related genes have shown relevance. A six-nucleotide deletion polymorphism (−652 6N del) in the promoter region of the CASP8 gene reduces caspase-8 expression and activity, thereby impairing the immune-mediated apoptosis of neoplastic cells. This variant is thought to attenuate the activation-induced cell death (AICD) of T lymphocytes, potentially weakening immune surveillance and enhancing the risk of malignant transformation [17]. Host genetic factors involved in DNA repair (e.g., GTF2H4, DUT, DMC1) and viral response or cell entry (e.g., OAS3, SULF1, IFNG) also significantly influence the persistence of HPV infection and the risk of progression to cervical neoplasia [19].

Beyond genetic predispositions, epigenetic dysregulation plays a crucial role in the transformation from LSILs to HSILs and invasive carcinoma. The promoter hypermethylation of tumor suppressor genes and signaling modulators is a hallmark of disease progression. For instance, the hypermethylation of CCNA1 and C13ORF18, both of which are key regulators of the cell cycle, is strongly associated with high-grade cervical intraepithelial neoplasia and cancer but is infrequent in normal or low-grade lesions [20]. Epigenetic silencing of the SFRP gene family, which negatively regulates the Wnt/β-catenin signaling pathway, leads to pathway activation and downstream effects such as the enhanced proliferation and evasion of apoptosis. This aberrant Wnt signaling further drives the progression of lesions along the LSIL–HSIL–carcinoma spectrum [21].

In addition, the cumulative methylation of multiple tumor suppressor genes—including DAPK, HIC-1, HIN-1, MGMT, RAR-β, RASSF1A, SHP-1, and the EMT-associated transcription factor Twist—has been shown to correlate with the increasing severity of cervical lesions. These methylation events impair apoptosis, DNA repair, and cell differentiation, and they facilitate epithelial-to-mesenchymal transition (EMT), a critical step for invasion and metastasis [24]. Collectively, these genetic and epigenetic alterations form a synergistic network that disrupts cellular homeostasis and immune surveillance, thereby promoting the initiation, persistence, and malignant transformation of cervical lesions.

Our model explicitly incorporates genetic polymorphisms and epigenetic DNA methylation as critical determinants influencing cervical cancer risk. Several SNPs, including Pre-miR-218 rs11134527, IFNG rs11177074, EVER1/EVER2 rs0903818, CASP8-652 6N del/ins, and combinations such as CD28+17 (rs3116496) and IFNG+874 (rs2430561), were identified as key genetic risk factors [16,17,18]. Additionally, specific SNPs linked to HPV persistence, including OAS3 rs12302655, SULF1 rs4737999, DUT rs3784621, and GTF2H4 rs2894054, were integrated into the risk model. DNA methylation patterns involving genes like CCNA1, C13ORF18, DAPK, HIC-1, HIN-1, MGMT, RAR-beta, and RASSF1A were also modeled as influential epigenetic factors affecting different transition states from normal tissue to invasive carcinoma [19,20,21,31].

The findings clearly illustrate the significant modulation of cervical cancer risk by these genetic biomarkers. Genetic and epigenetic variations markedly impact disease progression, particularly in HPV-infected women. For example, combinations of genetic polymorphisms such as Pre-miR-218 expression and the methylation of C13ORF18/DAPK substantially elevated cervical cancer risk compared to profiles involving LAMB3 alone. These genetic biomarkers also influenced risk in HPV-negative women, albeit to a lesser degree, underscoring the necessity of integrating genetic and epigenetic assessments alongside HPV status to refine cervical cancer screening strategies.

### 3.3. Cervical Cancer Precision Prevention Guided by Risk Score Percentile

Although Pap smear screening significantly reduces cervical cancer incidence, its efficacy has plateaued, indicating the need for the improved targeting of high-risk populations. By employing our proposed risk assessment model, we stratified women into distinct risk categories (low, intermediate, and high) based on risk score percentiles, corresponding to lifetime cervical cancer risks ranging from 0.11% to 9.53%. This stratification allows for precisely tailored preventive interventions.

Our results indicated that, for women classified as low risk (<20th percentile), longer intervals between Pap smears combined with HPV vaccination effectively managed cervical cancer risk. Conversely, intermediate (20th-80th percentile) and high-risk (>80th percentile) groups benefited significantly from intensified interventions, including frequent screenings coupled with HPV vaccination and HPV DNA testing. The marginal benefits of vaccination ranged from 3% to 18%, being notably greater among higher-risk groups, highlighting the crucial role of vaccination in reducing cervical cancer burden among the women most vulnerable to the threat of cervical cancer. The precision approach guided by risk stratification thus not only optimizes resource utilization but also potentially improves adherence by aligning preventive measures with individual risk perception. This method represents a significant advancement toward precision medicine in cervical cancer prevention, ultimately aiming to substantially reduce both incidence and mortality.

The HPV infection–Precancerous–Cancer model captures the dynamic evolution of cervical lesions while incorporating the modifying effects of genetic and epigenetic markers. Rather than conventional “training” using labeled datasets, the model functions as a mechanistic risk prediction framework that synthesizes published parameter estimates and biomarker-specific effect sizes to simulate disease trajectories under different intervention strategies. To simulate real clinical scenarios, the model was applied to six illustrative case profiles with varying molecular risk levels and HPV infection persistence. These representative cases demonstrated how the model could inform risk-stratified prevention strategies, comparing outcomes between universal triennial Pap smear screening and individualized approaches tailored to each woman’s molecular risk percentile.

In high- and intermediate-risk individuals, the model recommended more intensive preventive strategies—such as combined HPV vaccination, DNA testing, and Pap smear at shorter intervals. Conversely, in low-risk individuals, extended inter-screening intervals were found to maintain cancer risk at acceptable levels. These simulations underscore the model’s clinical utility in tailoring cervical cancer prevention based on molecular risk stratification. This modeling framework supports a transition from a one-size-fits-all screening approach to a precision prevention paradigm, with potential to enhance resource allocation, minimize over-screening in low-risk women, and improve early detection in high-risk groups.

While evaluating the cost-effectiveness of personalized precision screening is paramount, it is beyond the scope of the current study. Chen et al. (2011) [3] conducted a cost-effectiveness analysis of cervical cancer prevention strategies in Taiwan using a probabilistic Markov decision model. Their study concluded that annual Pap smear screening remains the most cost-effective approach, while combining HPV DNA testing with triennial screening is economically favorable under reasonable willingness-to-pay thresholds. Additionally, HPV vaccination coupled with triennial screening could become cost-effective if vaccine prices significantly decrease. Personalized screening tailors both the screening method and frequency to an individual’s specific risk profile. By integrating genetic and epigenetic biomarkers with conventional cytology and HPV testing, women at higher risk can receive more intensive surveillance. Such a risk-stratified strategy may maximize clinical benefits, enhance cost-effectiveness, reduce unnecessary interventions, and promote true precision in preventive care. This represents a crucial area for future research.

While genetic and epigenetic biomarkers hold significant promise for enabling personalized cervical cancer prevention, incorporating genetic testing into national screening programs presents important ethical and practical challenges. These challenges are best examined through the lens of the four core bioethical principles: autonomy, beneficence, non-maleficence, and justice. Respect for autonomy requires that individuals are fully informed and able to freely choose whether to undergo genetic testing. This entails rigorous informed consent procedures and transparent communication about the implications, limitations, and potential psychological impacts of receiving genetic risk information. Opt-in or opt-out mechanisms must be designed to protect individual choice and avoid coercion. Beneficence obliges the screening program to demonstrably improve health outcomes. Genetic testing should provide actionable insights—such as the identification of high-risk individuals who would benefit from intensified surveillance or early intervention—that lead to meaningful clinical benefits. Non-maleficence emphasizes the importance of minimizing harm. Potential risks include false positives, overdiagnosis, stigmatization, anxiety, and unnecessary interventions. To mitigate these harms, the program must ensure that test results are accompanied by high-quality genetic counseling and that thresholds for clinical action are clearly evidence-based. Justice demands fair access and the equitable distribution of benefits and burdens. Screening infrastructure must be designed to avoid exacerbating existing disparities based on socioeconomic status, geography, or digital literacy. Policymakers must also consider the affordability of follow-up care and the long-term psychosocial consequences for individuals labeled as genetically high-risk. Although technically feasible, the implementation of genetic testing within a population-wide program requires careful policy design, public engagement, and ethical safeguards. It must be coupled with health system readiness, workforce training in genetic counseling, data privacy protections, and the ongoing evaluation of clinical utility and social acceptability [32].

### 3.4. Limitations

This study has several limitations. The distributions and effect sizes of genetic and epigenetic biomarkers were primarily derived from the published literature and embedded into a virtual cohort reflective of the cervical cancer risk profile of Taiwanese women. We assumed independence among biomarkers and adopted a uniform joint distribution framework. For each simulated individual, the effects of genetic and epigenetic risk factors were incorporated into the model via a proportional-hazard structure, using additive log-linear transformations [25,33,34,35]. While this approach demonstrates the feasibility of implementing a personalized, risk-guided cervical cancer prevention strategy, it remains dependent on literature-based effect size estimates.

Although prior studies support the additive contribution of polygenes in cancer risk modeling [36], recent findings suggest that long-range genetic interactions may violate the additivity assumption [37]. Thus, the future integration of population-scale biobank data with our dynamic disease model could help reduce dependence on published estimates and enable the exploration of nonlinear and interactive effects among biomarkers through machine learning-based methods. Such integration represents a promising direction for expanding the precision prevention paradigm.

In addition, to partially address uncertainty in key inputs, we conducted a sensitivity analysis by varying the prevalence of HPV infection—one of the primary risk factors—from 12.7% to 5.1%. We evaluated the robustness of combined prevention strategies under this lower prevalence assumption. The results, presented in Appendix A, demonstrate that, while the overall trend in the effectiveness of prevention strategies remained consistent, the absolute impact was attenuated in the low-prevalence scenario—particularly for women with a risk score percentile below 40%. Notably, the protective effect of HPV vaccination in this subgroup was substantially diminished and did not reach statistical significance.

These findings reinforce the need to continuously refine the model using real-world data and underscore the importance of embedding empirical uncertainty into future risk predictions. They also highlight the potential utility of integrating biobank resources rich in molecular and clinical information to support adaptive, evidence-based cervical cancer prevention strategies.

## 4. Materials and Methods

### 4.1. Study Population

To evaluate the effects of HPV infection status and genetic and epigenetic biomarkers on the progression of cervical neoplasms from LSIL and HSIL to invasive carcinoma, we embedded data abstracted from the literature into a virtual cohort representative of the Taiwanese population [2,3]. In this study, the virtual cohort was designed to represent the Taiwanese female population by integrating demographic characteristics, HPV infection status, and the distribution of genetic and epigenetic biomarkers abstracted from published sources. We used natural history parameters of cervical neoplasms—such as transition probabilities from LSIL to HSIL and to invasive carcinoma—under conditions of no screening or vaccination, to simulate disease progression over time. Molecular profiles, including susceptibility genotypes and promoter methylation patterns, were embedded to reflect risk heterogeneity across individuals.

This virtual cohort allowed us to evaluate the potential impact of various cervical cancer prevention strategies, including HPV vaccination, HPV DNA testing, and Pap smear screening, applied individually or in combination at different intervals. These strategies were modeled within the cohort to estimate their effectiveness in preventing lesion progression and reducing cervical cancer burden. Importantly, this simulated population also served as a control group scenario—representing natural disease progression without intervention—against which the effects of each prevention strategy could be compared. Risk stratification was performed by assigning molecular risk scores derived from embedded biomarker profiles, enabling the assessment of precision prevention approaches across different risk groups.

The evolution of cervical cancer was projected using the virtual cohort consisting of 300,000 Taiwanese women aged 30 years or older, which was simulated to construct a multistate risk assessment model for cervical cancer. Before the implementation of the population-based Pap smear screening program, the incidence of cervical cancer was as high as 30 per 100,000 women. Epidemiological parameters, including an HPV infection rate of 0.127 and an HPV 16/18 type prevalence of 0.692, were obtained from previous studies [38,39,40,41,42,43] and used to stratify women into groups with and without HPV infection.

### 4.2. Review-Based and Empirical-Based Parameters

The effects of genetic markers, along with their proportions in the general population, on the development of cervical cancer lesions are summarized in Table 1. Three genetic loci (CD28+17(TT)INFG+874, Pre-mir-218 rs1113452, and CASP8-652 6N del), with relative ratios ranging from 0.53 to 0.86, demonstrated inverse associations with cervical carcinogenesis progressing from normal tissue to CIN1. In contrast, the LAMB3 rs2566 locus exhibited a 1.5-fold positive association with cervical carcinogenesis progressing from HPV infection to CIN1. Additionally, three loci (IFNG rs11177074, POLN rs17132382, and TMC8 rs9893818) showed relative ratios between 1.35 and 2.47, indicating positive associations with carcinogenesis from HPV infection to CIN1.

### 4.3. Derivation of Risk Scores Driving Cervical Cancer Evolution

#### 4.3.1. Genetic Biomarkers Associated with the Evolution of Cervical Cancer

The *i*th individual’s risk score given HPV status (j = 0 HPV negative, j = 1 HPV positive) was calculated using the formula:RiskScore_ij_ = log(λ₀_j_) + ∑(βᵢ × *x*ᵢ),
where λ₀_j_ represents baseline hazard rate given the *j*th HPV status, *x*ᵢ represents the genetic or epigenetic profile of a given woman, and βᵢ denotes the log-transformed effect size reported in prior studies (see Table 1 and Table 2). The coefficients corresponding to these factors were derived from the relative risks presented in Table 1 and Table 2. The probability of each woman having specific genetic and epigenetic factors was randomly assigned using a uniform distribution, based on the respective population proportions derived from the literature.

Risk scores were then transformed into percentiles within the simulated cohort, and both the raw scores and percentiles are presented in Table 3. For example, Case A’s risk score is computed as follows:(−2.00) + (−0.15) + (4.21) + (0.75) = 2.81, 
based on the following molecular features: HPV positivity, Pre-mir-218 rs1113452 (AA), C13ORF18 positivity, and DAPK positivity. Risk scores for the remaining five illustrative cases can be similarly derived.

#### 4.3.2. Projection of Computer Simulation

We simulated a hypothetical cohort incorporating the previously mentioned genotypes and DNA methylation proportions, mimicking those observed in the control group (women without cervical cancer), together with their relative magnitudes of risk for cervical cancer or persistent HPV infection. A birth cohort was separately simulated, in which each woman was followed from birth up to 79 years of age. Through this lifetime simulation, the development of LSILs, HSILs, and cervical cancer among these women could be predicted.

### 4.4. Model Validation

Internal validation was conducted using a simulated Taiwanese cohort with input parameters derived from national datasets and prior studies [2,3]. Key parameters included a cervical cancer incidence of 26.3 per 100,000 woman-years, an HPV prevalence of 12.7%, a 3-year screening interval, and a 52.2% attendance rate.

For external validation, the model was applied to an independent dataset reflecting the Japanese female population, using inputs from IOC/IARC (2023) [44] and Palmar et al. in 2024 [45]. These included a cervical cancer incidence of 19.9 per 100,000 woman-years, an HPV prevalence of 10.3%, a 2-year screening interval, and an attendance rate of 43.7%. Model-predicted cervical cancer cases were then compared with observed case estimates using a goodness-of-fit (χ^2^) test.

### 4.5. Precision Intervention Strategies

The paradigm of cervical cancer prevention is transitioning toward a precision approach, leveraging advances in molecular diagnostics. Genetic and epigenetic biomarkers, when integrated with HPV testing, enable more accurate stratification of women at elevated risk of progression to invasive disease [46,47]. The proposed HPV infection–Precancerous–Cancer model dynamically incorporates these risk determinants, allowing individualized risk quantification. This enables evidence-based recommendations for targeted prevention strategies, ensuring optimal resource allocation and reducing over-screening in low-risk populations. We presented outcomes regarding cervical cancer risk under various preventive strategies, including Pap smear screening, Pap smear screening combined with one-time HPV DNA testing, as well as HPV vaccination followed by Pap smear screening with and without HPV DNA testing. Additionally, individualized screening strategies with different screening intervals based on each woman’s risk score were also demonstrated.

## 5. Conclusions

The development of an HPV infection–Precancerous–Cancer risk assessment model incorporating genetic biomarkers, including genetic and epigenetic factors, facilitates health policymakers in designing individualized precision prevention strategies through various combinations of Pap smear screening, HPV DNA testing, and HPV vaccination.

## Figures and Tables

**Figure 1 ijms-26-06016-f001:**
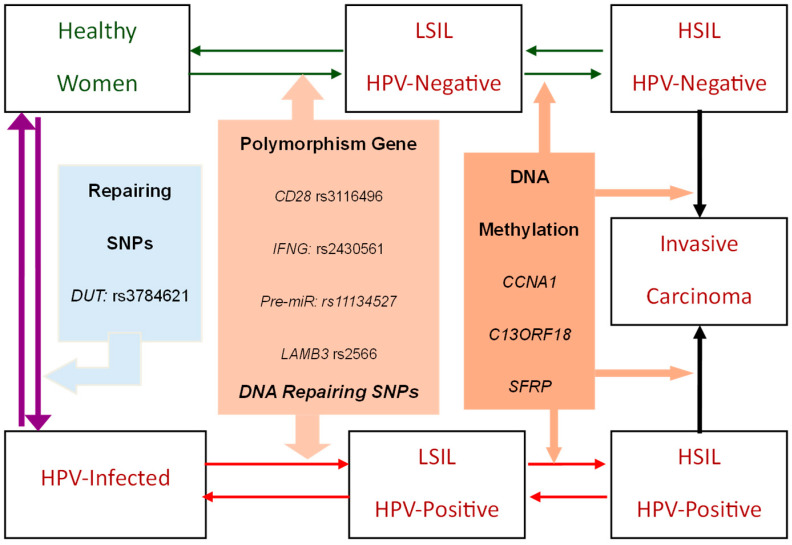
Evolution of cervical lesions to invasive carcinoma with the consideration of HPV infection status and the effect of genetic and epigenetic markers. The green arrows represent the transitions between healthy status and both LSIL and HSIL among HPV-negative individuals. The red arrows indicate the dynamic progression of HPV-positive individuals, illustrating transitions from HPV infection to LSIL and HSIL, as well as the subsequent progression to invasive carcinoma. The evolution of cervical neoplasia is modulated by SNPs and DNA methylation (orange arrows). The purple arrows denote the transitions between infection and healthy state, which is regulated by DNA repair-related SNPs.

**Figure 2 ijms-26-06016-f002:**
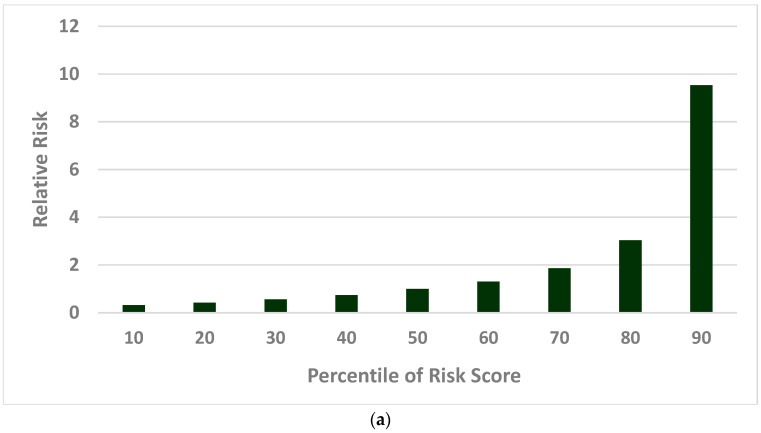
Cervical cancer risk by percentiles of risk score: (**a**) Relative risk of cervical cancer. (**b**) Lifetime risk of cervical cancer.

**Figure 3 ijms-26-06016-f003:**
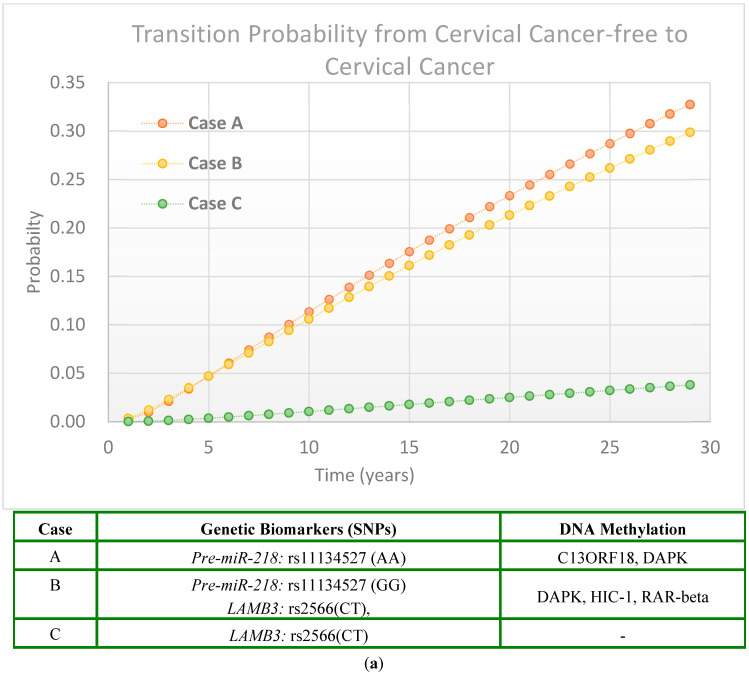
Illustrations of individual risk for transition from cervical cancer-free to cervical cancer for women with and without HPV infection: (**a**) Women with HPV infection. (**b**) Women without HPV infection.

**Table 1 ijms-26-06016-t001:** Genetic Biomarkers Related to the Incidence of LSIL and Persistent HPV Infection.

State Transition	Genetic Marker	RR	Population Proportion	Ref.
Normal → LSIL	CD28+17 (TT) INFG+874 (AA)	0.78	23.60%	[18]
	Pre-mir-218 rs11134527 (GG) (AA)	0.73 0.86	17.81% 47.55%	[16]
	LAMB3 rs2566 (TT) (CT)	1.80 1.59	10.19% 46.29%	[16]
	CASP8-652 6N del/del del/ins	0.53 0.75	10.27% 39.79%	[17]
	DUT rs3784621 (CC) (CT)	1.54 1.33	11.35% 43.50%	[19]
	GTF2H4 rs2894054 (AA) (AG)	0.11 0.57	2.12% 24.24%	[19]
	OAS3 rs12302655 (AA) (AG)	1.57 --	13.18% 0%	[19]
	SULF1 rs4737999 (AA) (AG)	0.59 0.59	7.76% 38.35%	[19]
HPV → LSIL	IFNG rs11177074 (CC) (CT)	1.35 1.78	0.25% 8.71%	
	POLN rs17132382 (TT) (CT)	2.47 2.16	3.88% 27.43%	[19]
	TMC8 rs9893818 (AA) TMC6 (AC)	1.57 -	5.10% 0%	

**Table 2 ijms-26-06016-t002:** Methylation effect on transition of cervical cancer lesion and cervical cancer.

Methylation	Population Frequency	Relative Risk for			Ref.
		Normal → LSIL	LSIL → HSIL	HSIL → Cancer	
CCNA1	4.85%	42.08	-	-	[21]
C13ORF18	2.91%	67.66	-	-	[21]
SFRP	4.44%	3.92	1.06	18.37	[22]
DAPK	26.83%	2.11	3.44	0.97	[24]
HIC-1	24.39%	2.72	1.66	0.99	[24]
HIN-1	9.76%	2.13	1.57	1.35	[24]
MGMT	2.44%	1.29	4.07	2.47	[24]
RAR-beta	4.88%	3.58	4.35	1.32	[24]
RASSF1A	4.88%	2.77	2.31	1.27	[24]
SHP-1	4.88%	6.38	1.42	1.02	[24]
Twist	7.32%	1.80	4.54	0.79	[24]

**Table 3 ijms-26-06016-t003:** Detailed profiles, derived risk scores, percentiles of risk scores, and the effectiveness of precision prevention guided by the risk percentile compared with the universal triennial Pap smear of the six illustrative cases.

Profiles	RR/ Baseline Hazard Rate of HPV (λ₀_j_)	Regression Coefficients (β)/ln(λ₀_j_)	Case A	Case B	Case C	Case D	Case E	Case F
CD28+17 (TT) INFG+874 (AA)	0.78	−0.25					V	V
Pre-mir-218 rs11134527 (GG)	0.73	−0.31						V
Pre-mir-218 rs11134527 (AA)	0.86	−0.15	V			V		
LAMB3 rs2566 (TT)	1.80	0.59				V		
LAMB3 rs2566 (CT)	1.59	0.46		V	V		V	
CASP8-652 6N del/del	0.53	−0.63						
CASP8-652 6N del/ins	0.75	−0.29						
DUT rs3784621 (CC)	1.54	0.43				V		
DUT rs3784621 (CT)	1.33	0.29					V	
GTF2H4 rs2894054 (AA)	0.11	−2.21						
GTF2H4 rs2894054 (AG)	0.57	−0.56						
OAS3 rs12302655 (AA)	1.57	0.45						
SULF1 rs4737999 (AA)	0.59	−0.53						
SULF1 rs4737999(AG)	0.59	−0.53						
IFNG rs11177074 (CC)	1.35	0.30						
IFNG rs11177074 (CT)	1.78							
POLN rs17132382 (CT)	2.16	0.77						
POLN rs17132382 (TT)	2.47	0.90				V		
TMC8 rs9893818 (AA)	1.57	0.45						
CCNA1	42.08	3.74						
C13ORF18	67.66	4.21	V					
SFRP	3.92	1.37					V	
DAPK	2.11	0.75	V	V		V	V	
HIC-1	2.72	1.00		V		V		
HIN-1	2.13	0.76						
MGMT	1.29	0.25						
RAR-beta	3.58	1.28		V				
RASSF1A	2.77	1.02						
SHP-1	6.38	1.85						
Twist	1.80	0.59						
**Persistent HPV infection Status**	λ₀_j_ Positive (j = 1): 0.135 Negative (j = 0): 0.051	ln(λ₀_j_) Positive (j = 1): −2.00 Negative (j = 0): −2.98	Positive	Positive	Positive	Negative	Negative	Negative
**Risk Score (log(λ_0j_)+** $\sum \beta_{i}x_{i}$ **)**	-	-	2.81	1.16	−1.55	0.57	−0.36	−3.53
**Percentile of Risk**	-	-	>80%	>80%	40–60%	60–80%	60–80	<20
**RR (vs. Triennial Pap Smear)**	-	-	0.61 (0.53–0.71) ^1^ *p* < 0.001	0.61 (0.53–0.71) ^1^ *p* < 0.001	0.83 (0.71–0.97) ^2^ *p* = 0.02	0.55 (0.48, 0.64) ^3^ *p* < 0.001	0.55 (0.48, 0.64) ^3^ *p* < 0.001	1.02 (0.71, 1.46) ^4^ *p* = 0.93

^1^ HPV vaccination + HPV test + Annual Pap smear; ^2^ HPV test + Triennial Pap smear; ^3^ HPV vaccination + Triennial Pap smear; ^4^ Pap smear at 6-year interval.

**Table 4 ijms-26-06016-t004:** Simulated relative risk of cervical cancer by precision prevention strategies *.

	Relative Risk (95% CI)									
Risk Score Percentile	<20%		20–40%		40–60%		60–80%		>80%	
Prevention Strategy	HPV Testing		HPV Testing		HPV Testing		HPV Testing		HPV Testing	
	−	+	−	+	−	+	−	+	−	+
**Pap Smear Screening by Inter-screening Interval**										
**1 yr**	0.37	0.31	0.33	0.28	0.35	0.35	0.44	0.40	0.42	0.44
	(0.28,0.49)	(0.28,0.39)	(0.31,0.39)	(0.40,0.48)	(0.38,0.47)	(0.23,0.41)	(0.24,0.34)	(0.31,0.39)	(0.36,0.44)	(0.40,0.48)
**3 yr**	0.45	0.35	0.35	0.33	0.40	0.33	0.46	0.46	0.51	0.45
	(0.34,0.58)	(0.30,0.41)	(0.36,0.45)	(0.42,0.51)	(0.47,0.56)	(0.27,0.47)	(0.28,0.39)	(0.29,0.37)	(0.42,0.50)	(0.41,0.50)
**5 yr**	0.44	0.42	0.43	0.41	0.42	0.39	0.50	0.46	0.54	0.48
	(0.34,0.57)	(0.37,0.49)	(0.38,0.47)	(0.46,0.54)	(0.50,0.60)	(0.33,0.54)	(0.36,0.48)	(0.35,0.43)	(0.42,0.50)	(0.44,0.53)
**HPV Vaccination + Pap Smear Screening by Inter-screening Interval**										
**1 yr**	0.32	0.28	0.15	0.19	0.21	0.20	0.24	0.23	0.29	0.30
	(0.23,0.46)	(0.20,0.40)	(0.11,0.19)	(0.15,0.24)	(0.18,0.25)	(0.17,0.24)	(0.21,0.27)	(0.20,0.27)	(0.25,0.33)	(0.27,0.35)
**3 yr**	0.32	0.29	0.24	0.21	0.26	0.23	0.26	0.25	0.36	0.32
	(0.23,0.46)	(0.20,0.41)	(0.19,0.30)	(0.17,0.27)	(0.23,0.31)	(0.20,0.27)	(0.22,0.29)	(0.22,0.29)	(0.32,0.41)	(0.28,0.36)
**5 yr**	0.39	0.34	0.27	0.20	0.25	0.24	0.28	0.26	0.36	0.32
	(0.28,0.54)	(0.25,0.48)	(0.22,0.33)	(0.16,0.25)	(0.21,0.29)	(0.21,0.28)	(0.24,0.32)	(0.23,0.30)	(0.32,0.41)	(0.28,0.37)
**HPV Vaccination**										
	0.94	0.96	0.55	0.61	0.57	0.58	0.64	0.58	0.73	0.70
	(0.73,1.20)	(0.76,1.21)	(0.47,0.65)	(0.52,0.71)	(0.51,0.63)	(0.52,0.65)	(0.58,0.70)	(0.53,0.64)	(0.66,0.81)	(0.63,0.77)
**Marginal effect of Vaccination**	3–8%		8–12%		10–17%		17–22%		13–18%	

* Simulated cohort with 300,000 Taiwan women, age starting from 0. Baseline: lacking any intervention, RR = 1.00.

## Data Availability

Data supporting this study are available from the corresponding author upon reasonable request.

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
