# Peer review of "Genetic Biomarkers Associated with Dynamic Transitions of Human Papillomavirus (HPV) Infection–Precancerous–Cancer of Cervix for Navigating Precision Prevention"

_ijms, 2025, doi:10.3390/ijms26136016_

Round 1
Reviewer 1 Report
Comments and Suggestions for Authors
This manuscript intends to develop a risk assessment model that incorporate genetic and epigenetic markers. The following are recommendations:
1. Describe how the 9 cases were generated and indicate the algorithm used in this process.
2. Please show if there is significant differences (i.e., p values) in the relative risk based on prevention strategies and among the 9 cases based on genetic markers.
3. When presenting relative risk, the 95% confidence interval must also be indicated.
4. The aims of this study are not clearly indicated. Hence, the results are presented in a confusing way. Is there correlation between prevention strategies and genetic/epigenetic changes? What is the clinical and practical use of this mathematical model?
5. Please provide a brief discussion on the pathophysiology of the mentioned genetic and epigenetic changes in the development/reversal of cervical lesions.
6. Please define what a "virtual cohort" means.
7. Indicate how the model was trained and tested for real clinical scenario.
8. Correct multiple grammatical errors.
Author Response
Please see attached PDF file for the full point-by-point response with formatted figures and tables.
Q1. Describe how the 9 cases were generated and indicate the algorithm used in this process.
Ans: Using the population distribution and effect sizes of relevant genetic and epigenetic markers, we generated a virtual cohort of 300,000 women that simulated the natural progression from normal cervical epithelium through LSIL and HSIL to invasive cervical carcinoma. To assess the influence of HPV infection status, genetic polymorphisms, and DNA methylation markers on the progression of cervical neoplasia, we incorporated molecular data from the literature into the model. This virtual cohort was constructed to be representative of the Taiwanese population. Natural history parameters reported by Chen et al. (2011) were used to estimate disease progression in the absence of preventive interventions, incorporating individual-level molecular profiles.
To illustrate this dynamic transition from HPV infection to precancerous lesions and finally to cervical cancer, six representative cases were selected (Figure 3, Cases A to F). The first three cases (Cases A, B, and C; Figure 3a) represent women with HPV infection, while the remaining three (Cases D, E, and F; Figure 3b) are HPV-negative. The complete genetic and DNA methylation profiles of these six individuals are summarized in the profile table.
The ith individual’s risk score given HPV status (j=0 HPV negative, j=1 HPV positive) was calculated using the formula:
RiskScoreij = log(λ₀j) + ∑(βᵢ × xᵢ)
where λ₀j represents baseline hazard rate given the jth HPV status, xᵢ represents the genetic or epigenetic profile of a given woman, βᵢ denotes the log-transformed effect size reported in prior studies (see Table 1). Risk scores were then transformed into percentiles within the simulated cohort, and both the raw scores and percentiles are presented in Table 3.
For example, Case A’s risk score is computed as:
(−2.00) + (−0.15) +(4.21)+ (0.75) = 2.81,
based on the following molecular features: HPV positivity, Pre-mir-218 rs1113452 (AA), C13ORF18 positivity and DAPK positivity.
Table 3. Detailed profiles and derived risk scores, percentile of risk scores, and the effectiveness of precision prevention guided by the risk percentile compared with universal triennial Pap smear of the six illustrative cases
Profiles |
RR / Baseline Hazard Rate of HPV (λ₀j) |
Regression Coefficients (β)/ ln(λ₀j) |
Case A |
Case B |
Case C |
Case D |
Case E |
Case F |
CD28+17 (TT) INFG+874 (AA) |
0.78 |
-0.25 |
|
|
|
|
V |
V |
Pre-mir-218 rs11134527 (GG) |
0.73 |
-0.31 |
|
V |
|
|
|
V |
Pre-mir-218 rs11134527 (AA) |
0.86 |
-0.15 |
V |
|
|
V |
|
|
LAMB3 rs2566 (TT) |
1.80 |
0.59 |
|
|
|
V |
|
|
LAMB3 rs2566 (CT) |
1.59 |
0.46 |
|
V |
V |
|
V |
|
CASP8 -652 6N del/del |
0.53 |
-0.63 |
|
|
|
|
|
|
CASP8 -652 6N del/ins |
0.75 |
-0.29 |
|
|
|
|
|
|
DUT rs3784621 (CC) |
1.54 |
0.43 |
|
|
|
V |
|
|
DUT rs3784621 (CT) |
1.33 |
0.29 |
|
|
|
|
V |
|
GTF2H4 rs2894054 (AA) |
0.11 |
-2.21 |
|
|
|
|
|
|
GTF2H4 rs2894054 (AG) |
0.57 |
-0.56 |
|
|
|
|
|
|
OAS3 rs12302655 (AA) |
1.57 |
0.45 |
|
|
|
|
|
|
SULF1 rs4737999 (AA) |
0.59 |
-0.53 |
|
|
|
|
|
|
SULF1 rs4737999(AG) |
0.59 |
-0.53 |
|
|
|
|
|
|
IFNG rs11177074 (CC) |
1.35 |
0.30 |
|
|
|
|
|
|
IFNG rs11177074 (CT) |
1.78 |
|
|
|
|
|
|
|
POLN rs17132382 (CT) |
2.16 |
0.77 |
|
|
|
|
|
|
POLN rs17132382 (TT) |
2.47 |
0.90 |
|
|
|
V |
|
|
TMC8 rs9893818 (AA) |
1.57 |
0.45 |
|
|
|
|
|
|
CCNA1 |
42.08 |
3.74 |
|
|
|
|
|
|
C13ORF18 |
67.66 |
4.21 |
V |
|
|
|
|
|
SFRP |
3.92 |
1.37 |
|
|
|
|
V |
|
DAPK |
2.11 |
0.75 |
V |
V |
|
V |
V |
|
HIC-1 |
2.72 |
1.00 |
|
V |
|
V |
|
|
HIN-1 |
2.13 |
0.76 |
|
|
|
|
|
|
MGMT |
1.29 |
0.25 |
|
|
|
|
|
|
RAR-beta |
3.58 |
1.28 |
|
V |
|
|
|
|
RASSF1A |
2.77 |
1.02 |
|
|
|
|
|
|
SHP-1 |
6.38 |
1.85 |
|
|
|
|
|
|
Twist |
1.80 |
0.59 |
|
|
|
|
|
|
Persistent HPV infection status |
λ₀j Positive (j=1): 0.135 Negative (j=0): 0.051 |
ln(λ₀j) Positive (j=1): -2.00 Negative (j=0): -2.98 |
Positive |
Positive |
Positive |
Negative |
Negative |
Negative |
Risk Score (log(λ0j)+ ) |
- |
- |
2.81 |
1.16 |
-1.55 |
0.57 |
-0.36 |
-3.53 |
Percentile of Risk |
- |
- |
>80% |
>80% |
40-60% |
60-80% |
60-80 |
<20 |
RR (vs Triennial Pap Smear) |
- |
- |
0.61 (0.53-0.71)1 p<0.001 |
0.61 (0.53-0.71)1 p<0.001 |
0.83 (0.71-0.97)2 P=0.02 |
0.55 (0.48, 0.64)3 p < 0.001 |
0.55 (0.48, 0.64)3 p < 0.001 |
1.02 (0.71, 1.46)4 p = 0.93 |
1 HPV vaccination + HPV test + Annual Pap smear
2 HPV test + Triennial Pap smear
3 HPV vaccination + Triennial Pap smear
4 Pap smear at 6-year interval
To facilitate interpretation, we also provide distributions of the overall risk score, as well as stratified distributions for HPV-positive and HPV-negative individuals, in Supplementary Figures S1 and S2.
All derivations for these six illustrative cases, including detailed risk profiles, corresponding risk scores, and percentiles, are comprehensively documented in the revised manuscript (Sections 2.2.3 Scenarios of Personalized Risk Assessment, 4.3.1 Genetic biomarkers associated with the evolution of Cervical Cancer, and the Supplementary Materials)
Supplementary Figure S1. Distribution of overall risk score and the risk score of HPV positive and HPV negative women.
Supplementary Figure S2. Distribution of overall risk score and the risk score of HPV-positive (a) and HPV-negative (b) women along with the risk sores of six illustrative cases
- Women with positive HPV infection (Case A, Case B, and Case C)
- Women with positive HPV infection (Case D, Case E, and Case F)
Reference
Chen MK, et al.: Cost-effectiveness analysis for Pap smear screening and human papillomavirus DNA testing and vaccination. J Eval Clin Pract. 2011;17(6):1050-1058.
Q2. Please show if there is significant differences (i.e., p values) in the relative risk based on prevention strategies and among the 9 cases based on genetic markers.
Ans. Based on the individual risk percentiles derived from molecular profiles, the relative effectiveness of various cervical cancer prevention strategies was evaluated across six illustrative cases. Cases A and B, both exceeding the 80th risk percentile, were classified as high-risk. For these cases, the combination of HPV vaccination, HPV testing, and annual Pap smears resulted in a significant risk reduction of 39% compared to the standard triennial Pap smear strategy (Relative Risk [RR]: 0.61; 95% Confidence Interval [CI]: 0.53–0.71; p < 0.001).
For women at intermediate risk levels—represented by Case C (40-60th percentile), Case D, and Case E (both 60–80th percentile)—alternative prevention strategies combining either HPV testing with triennial Pap smear or HPV vaccination with triennial Pap smear also demonstrated statistically significant risk reductions. Detailed relative risks and confidence intervals for these strategies are reported in Table 3 of the revised manuscript.
In contrast, for low-risk individuals such as Case F (<20th percentile), extending the inter-screening interval to six years yielded a comparable level of protection to the standard triennial Pap smear (RR: 1.02; 95% CI: 0.71-1.46; p = 0.93), indicating no significant increase in risk.
These findings support the feasibility of tailoring cervical cancer prevention strategies based on molecular risk stratification. The detailed statistical comparisons, including p values and confidence intervals, are summarized in Table 3 and discussed in Section 2.3 of the revised manuscript.
Q3. When presenting relative risk, the 95% confidence interval must also be indicated.
Ans. The 95% CIs of cervical cancer prevention strategies across the stratum of risk percentiles (Table 4) have been included in the relative risk table as follows.
Table 4. Simulated relative risk of cervical cancer by precision prevention strategies*
|
Relative Risk (95% CI) |
|
|||||||||||||||
Risk Score Percentile |
<20% |
20-40% |
40-60% |
60-80% |
>80% |
|
|||||||||||
Prevention Strategy |
HPV testing |
HPV testing |
HPV testing |
HPV testing |
HPV testing |
|
|||||||||||
- |
+ |
- |
+ |
- |
+ |
- |
+ |
- |
+ |
|
|||||||
Pap Smear Screening by Inter-screening Interval |
|
||||||||||||||||
1 yr |
0.37 |
0.31 |
0.33 |
0.28 |
0.35 |
0.35 |
0.44 |
0.40 |
0.42 |
0.44 |
|
||||||
(0.28,0.49) |
(0.28,0.39) |
(0.31,0.39) |
(0.40,0.48) |
(0.38,0.47) |
(0.23,0.41) |
(0.24,0.34) |
(0.31,0.39) |
(0.36,0.44) |
(0.40,0.48) |
|
|||||||
3 yr |
0.45 |
0.35 |
0.35 |
0.33 |
0.40 |
0.33 |
0.46 |
0.46 |
0.51 |
0.45 |
|
||||||
(0.34,0.58) |
(0.30,0.41) |
(0.36,0.45) |
(0.42,0.51) |
(0.47, 0.56) |
(0.27,0.47) |
(0.28, 0.39) |
(0.29,0.37) |
(0.42, 0.50) |
(0.41,0.50) |
|
|||||||
5 yr |
0.44 |
0.42 |
0.43 |
0.41 |
0.42 |
0.39 |
0.50 |
0.46 |
0.54 |
0.48 |
|
||||||
(0.34,0.57) |
(0.37,0.49) |
(0.38,0.47) |
(0.46,0.54) |
(0.50, 0.60) |
(0.33,0.54) |
(0.36, 0.48) |
(0.35,0.43) |
(0.42, 0.50) |
(0.44,0.53) |
|
|||||||
HPV Vaccination + Pap Smear Screening by Inter-screening Interval |
|
||||||||||||||||
1 yr |
0.32 |
0.28 |
0.15 |
0.19 |
0.21 |
0.20 |
0.24 |
0.23 |
0.29 |
0.30 |
|||||||
(0.23,0.46) |
(0.20, .40) |
(0.11,0.19) |
(0.15, 0.24) |
(0.18,0.25) |
(0.17, 0.24) |
(0.21,0.27) |
(0.20, 0.27) |
(0.25,0.33) |
(0.27, 0.35) |
||||||||
3 yr |
0.32 |
0.29 |
0.24 |
0.21 |
0.26 |
0.23 |
0.26 |
0.25 |
0.36 |
0.32 |
|||||||
(0.23,0.46) |
(0.20,0.41) |
(0.19,0.30) |
(0.17, 0.27) |
(0.23,0.31) |
(0.20, 0.27) |
(0.22,0.29) |
(0.22, 0.29) |
(0.32,0.41) |
(0.28, 0.36) |
||||||||
5 yr |
0.39 |
0.34 |
0.27 |
0.20 |
0.25 |
0.24 |
0.28 |
0.26 |
0.36 |
0.32 |
|||||||
(0.28,0.54) |
(0.25,0.48) |
(0.22,0.33) |
(0.16, 0.25) |
(0.21,0.29) |
(0.21, 0.28) |
(0.24,0.32) |
(0.23, 0.30) |
(0.32,0.41) |
(0.28, 0.37) |
||||||||
HPV Vaccination |
|
||||||||||||||||
|
0.94 |
0.96 |
0.55 |
0.61 |
0.57 |
0.58 |
0.64 |
0.58 |
0.73 |
0.70 |
|||||||
|
(0.73,1.20) |
(0.76,1.21) |
(0.47,0.65) |
(0.52, 0.71) |
(0.51,0.63) |
(0.52, 0.65) |
(0.58,0.70) |
(0.53, 0.64) |
(0.66,0.81) |
(0.63, 0.77) |
|||||||
Marginal effect of Vaccination |
3-8% |
8-12% |
10-17% |
17-22% |
13-18% |
|
|||||||||||
*Simulated Cohort with 300,000 Taiwan women age started from 0,
Baseline: lacking of any intervention, RR=1.00
Q4. The aims of this study are not clearly indicated. Hence, the results are presented in a confusing way. Is there correlation between prevention strategies and genetic/epigenetic changes? What is the clinical and practical use of this mathematical model?
Ans: The aims of this study are now clearly stated in the final paragraph of the revised Introduction:
“This study aims to develop a mathematical model that integrates HPV infection status, genetic alterations, and epigenetic biomarkers to simulate the natural history of cervical neoplasia. The model provides a foundation for designing precision prevention strategies that combine Pap smear screening, HPV DNA testing, and HPV vaccination.”
To elaborate, the study was conducted in three steps:
- Characterizing Risk Determinants: We first examined the role of HPV infection status along with genetic and epigenetic markers in influencing the progression from LSIL and HSIL to invasive cervical cancer.
- Modeling Disease Evolution: These molecular factors were then embedded into a virtual cohort, representative of Taiwanese women, to simulate lesion progression using published natural history parameters (Koong et al., 2006; Chen et al., 2011).
- Evaluating Prevention Strategies: Finally, the effectiveness of different combinations of prevention modalities—HPV vaccination, HPV testing, and Pap smears at varied intervals—was assessed across distinct molecular risk profiles.
The relationship between genetic/epigenetic changes and prevention strategies is mediated through their impact on individual risk percentiles. Women with high-risk molecular profiles benefit more from intensive prevention (e.g., combined HPV testing, vaccination, and annual Pap smear), while those at low risk may safely extend screening intervals. These stratified outcomes, based on molecular risk, are presented in Section 2.3 and Table 3 of the manuscript.
To emphasize clinical and practical utility, we have added the following content to Section 4.5 (Precision Intervention Strategies):
“The paradigm of cervical cancer prevention is transitioning toward a precision approach, leveraging advances in molecular diagnostics. Genetic and epigenetic biomarkers, when integrated with HPV testing, enable more accurate stratification of women at elevated risk of progression to invasive disease (Bedell et al., 2020; Garg et al., 2024). The proposed HPV infection–precancer–cancer model dynamically incorporates these risk determinants, allowing individualized risk quantification. This enables evidence-based recommendations for targeted prevention strategies, ensuring optimal resource allocation and reducing over-screening in low-risk populations.”
References
Koong SL et al.: Efficacy and cost-effectiveness of nationwide cervical cancer screening in Taiwan. J Med Screen. 2006;13 Suppl 1:S44-47.
Chen MK, et al.: Cost-effectiveness analysis for Pap smear screening and human papillomavirus DNA testing and vaccination. J Eval Clin Pract. 2011;17(6):1050-1058.
Bedell SL et al.: Cervical cancer screening: past, present, and future. Sexual medicine reviews 8.1 (2020): 28-37.
Garg P et al.: Emerging biomarkers and molecular targets for precision medicine in cervical cancer. Biochimica et Biophysica Acta (BBA)-Reviews on Cancer (2024): 189106.
Q5. Please provide a brief discussion on the pathophysiology of the mentioned genetic and epigenetic changes in the development/reversal of cervical lesions.
Ans. A brief discussion regarding the mechanisms of genetic and epigenic biomarker on the occurrence of cervical lesion has been incorporated into the revised article (3.2 Genetic and Epigenetic Molecular Biomarkers on the Risk of Cervical Cancer).
“Genetic susceptibility to cervical low-grade squamous intraepithelial lesions (LSIL) and their subsequent progression to high-grade lesions (HSIL) and invasive cervical cancer involves complex immunological, molecular, and cellular mechanisms. Notably, immunoregulatory genes located on chromosome 2q33 have been identified as significant contributors to cervical cancer susceptibility. Specific polymorphisms, such as CD28 +17 (TT) and IFNG +874 (AA), have been associated with increased risk of LSIL, underscoring the role of host immune modulation in early lesion development (Ivansson, Juko-Pecirep, & Gyllensten, 2010).
MicroRNA-related pathways also contribute to cervical carcinogenesis. The interaction between miR-218 and LAMB3 has been implicated in lesion progression. Specifically, the HPV-16 E6 oncoprotein downregulates miR-218, leading to the overexpression of LAMB3, which in turn facilitates HPV persistence and promotes the development of precancerous lesions such as LSIL (Zhou et al., 2010).
Furthermore, genetic variants in apoptosis-related genes have shown relevance. A six-nucleotide deletion polymorphism (-652 6N del) in the promoter region of the CASP8 gene reduces caspase-8 expression and activity, thereby impairing immune-mediated apoptosis of neoplastic cells. This variant is thought to attenuate activation-induced cell death (AICD) of T lymphocytes, potentially weakening immune surveillance and enhancing the risk of malignant transformation (Sun et al., 2007).
Host genetic factors involved in DNA repair (e.g., GTF2H4, DUT, DMC1) and viral response or cell entry (e.g., OAS3, SULF1, IFNG) also significantly influence the persistence of HPV infection and the risk of progression to cervical neoplasia (Wang et al., 2010).”
“Beyond genetic predispositions, epigenetic dysregulation plays a crucial role in the transformation from LSIL to HSIL and invasive carcinoma. Promoter hypermethylation of tumor suppressor genes and signaling modulators is a hallmark of disease progression. For instance, hypermethylation of CCNA1 and C13ORF18, both of which are key regulators of the cell cycle, is strongly associated with high-grade cervical intraepithelial neoplasia and cancer, but is infrequent in normal or low-grade lesions (Yang et al., 2009).
Epigenetic silencing of the SFRP gene family, which negatively regulates the Wnt/β-catenin signaling pathway, leads to pathway activation and downstream effects such as enhanced proliferation and evasion of apoptosis. This aberrant Wnt signaling further drives the progression of lesions along the LSIL–HSIL–carcinoma spectrum (Chung et al., 2009).
In addition, the cumulative methylation of multiple tumor suppressor genes—including DAPK, HIC-1, HIN-1, MGMT, RAR-β, RASSF1A, SHP-1, and the EMT-associated transcription factor Twist—has been shown to correlate with increasing severity of cervical lesions. These methylation events impair apoptosis, DNA repair, cell differentiation, and facilitate epithelial-to-mesenchymal transition (EMT), a critical step for invasion and metastasis (Kim et al., 2010).
Collectively, these genetic and epigenetic alterations form a synergistic network that disrupts cellular homeostasis and immune surveillance, thereby promoting the initiation, persistence, and malignant transformation of cervical lesions.”
References
Ivansson EL, Juko-Pecirep I, Gyllensten UB. Interaction of immunological genes on chromosome 2q33 and IFNG in susceptibility to cervical cancer. Gynecol Oncol. 2010;116(3):544-548.
Kim JH, Choi YD, Lee JS, Lee JH, Nam JH, Choi C. Assessment of DNA methylation for the detection of cervical neo-plasia in liquid-based cytology specimens. Gynecol Oncol. 2010;116(1):99-104. doi:10.1016/j.ygyno.2009.09.032
Sun T, Gao Y, Tan W, et al. A six-nucleotide insertion-deletion polymorphism in the CASP8 promoter is associated with susceptibility to multiple cancers. Nat Genet. 2007;39(5):605-613.
Wang SS, Gonzalez P, Yu K, et al. Common genetic variants and risk for HPV persistence and progression to cervical cancer. PloS One. 2010;5(1):e8667.
Yang N, Eijsink JJH, Lendvai A, et al. Methylation markers for CCNA1 and C13ORF18 are strongly associated with high-grade cervical intraepithelial neoplasia and cervical cancer in cervical scrapings. Cancer Epidemiol Biomark Prev Publ Am Assoc Cancer Res Cosponsored Am Soc Prev Oncol. 2009;18(11):3000-3007.
Zhou X, Chen X, Hu L, et al. Polymorphisms involved in the miR-218-LAMB3 pathway and susceptibility of cervical cancer, a case-control study in Chinese women. Gynecol Oncol. 2010;117(2):287-290.
Q6. Please define what a "virtual cohort" means.
Ans. A virtual cohort refers to a simulated population constructed using statistical and biological information derived from real-world data, literature, and established disease progression models. In this study, the virtual cohort was designed to represent the Taiwanese female population by integrating demographic characteristics, HPV infection status, and the distribution of genetic and epigenetic biomarkers abstracted from published sources. We used natural history parameters of cervical neoplasms—such as transition probabilities from LSIL to HSIL and to invasive carcinoma—under conditions of no screening or vaccination, to simulate disease progression over time. Molecular profiles, including susceptibility genotypes and promoter methylation patterns, were embedded to reflect risk heterogeneity across individuals.
This virtual cohort allowed us to evaluate the potential impact of various cervical cancer prevention strategies, including HPV vaccination, HPV DNA testing, and Pap smear screening, applied individually or in combination at different intervals. These strategies were modeled within the cohort to estimate their effectiveness in preventing lesion progression and reducing cervical cancer burden.
Importantly, this simulated population also served as a control group scenario—representing natural disease progression without intervention—against which the effects of each prevention strategy could be compared. Risk stratification was performed by assigning molecular risk scores derived from embedded biomarker profiles, enabling the assessment of precision prevention approaches across different risk groups.
This description has been incorporated into Section 4.1 (Study Population) of the revised manuscript.
Reference
Chen MK, et al.: Cost-effectiveness analysis for Pap smear screening and human papillomavirus DNA testing and vaccination. J Eval Clin Pract. 2011;17(6):1050-1058.
Q7. Indicate how the model was trained and tested for real clinical scenario.
Ans.
As elaborated in the response to Q6, the model was developed using a virtual cohort of 300,000 simulated Taiwanese women. This cohort was constructed by embedding population-based distributions of HPV infection, genetic susceptibility, and epigenetic biomarkers associated with cervical neoplasia. Transition probabilities between cervical health states—from normal epithelium to LSIL, HSIL, and invasive carcinoma—were informed by natural history data (Chen et al., 2011) and stratified according to molecular risk profiles.
The HPV infection–precancer–cancer model captures the dynamic evolution of cervical lesions while incorporating the modifying effects of genetic and epigenetic markers. Rather than conventional "training" using labeled datasets, the model functions as a mechanistic risk prediction framework that synthesizes published parameter estimates and biomarker-specific effect sizes to simulate disease trajectories under different intervention strategies.
To simulate real clinical scenarios, the model was applied to six illustrative case profiles with varying molecular risk levels and HPV infection persistence. These representative cases demonstrated how the model could inform risk-stratified prevention strategies, comparing outcomes between universal triennial Pap smear screening and individualized approaches tailored to each woman’s molecular risk percentile.
In high- and intermediate-risk individuals, the model recommended more intensive preventive strategies—such as combined HPV vaccination, DNA testing, and Pap smear at shorter intervals. Conversely, in low-risk individuals, extended inter-screening intervals were found to maintain cancer risk at acceptable levels. These simulations underscore the model's clinical utility in tailoring cervical cancer prevention based on molecular risk stratification.
This modeling framework supports a transition from a one-size-fits-all screening approach to a precision prevention paradigm, with potential to enhance resource allocation, minimize over-screening in low-risk women, and improve early detection in high-risk groups.
This approach and its clinical relevance are detailed in Section 3.3 (Cervical Cancer Precision Prevention Guided by Risk Score Percentile) of the revised manuscript.
Q8. Correct multiple grammatical errors.
Ans. The revised manuscript has been thoroughly reviewed to correct grammatical errors and improve readability.

Reviewer 2 Report
Comments and Suggestions for Authors
The integration of genetic and epigenetic biomarkers into a multistate model for cervical cancer progression is innovative and highly relevant for precision prevention. The modeling of dynamic transitions (including regression) reflects clinical realities better than traditional linear models. The simulation approach using a virtual cohort of 300,000 women is commendable and provides a scalable framework.
- While the model is novel, it is not externally validated against real-world longitudinal cohort data.
- The choice and weighting of biomarkers rely heavily on previously published effect sizes but lack sensitivity analyses to address uncertainty.
Clarity & Structure: The manuscript is mostly well-organized, with clear progression from background to results and discussion. Tables and figures (e.g., Figure 1, Table 1/2/3) effectively illustrate the key results.
- The abstract and some text suffer from awkward phrasing (e.g., “disease volution” → “disease evolution”).
- Several grammatical errors and redundancies hinder readability (e.g., “remarkable higher probability” should be “markedly higher”; "Nature history" → "Natural history").
- Section transitions (e.g., between Results 2.1 and 2.2) could be smoother to improve logical flow.
Methodological Rigor: The multistate model design, including backward transitions, is methodologically strong. Incorporation of relative risk and population frequency in the risk score formulation is robust.
- No mention of model calibration or internal validation.
- The assumption of uniform distribution for SNPs and methylation patterns, based on literature estimates, may not reflect population heterogeneity accurately.
- The equation for risk score (Formula 1) lacks clarity in explaining λ₀ and β parameters; also, it doesn't justify why additive log-linear modeling was chosen.
-
Expand the method section to include:
- Model validation/calibration procedures.
- Sensitivity analyses biomarker prevalence and RR inputs.
- Discuss the ethical implications and feasibility of genetic testing in national screening programs.
- Compare existing models to contextualize innovation and impact.
- Consider including cost-effectiveness projections for the precision prevention strategies proposed.
Recommendations for Improvement: Clarify language throughout the manuscript and ensure professional editing.
Author Response
Please see attached PDF file for the point-by-point response with fomatted figures and tables.
Comments and Suggestions for Authors
The integration of genetic and epigenetic biomarkers into a multistate model for cervical cancer progression is innovative and highly relevant for precision prevention. The modeling of dynamic transitions (including regression) reflects clinical realities better than traditional linear models. The simulation approach using a virtual cohort of 300,000 women is commendable and provides a scalable framework.
Q1. While the model is novel, it is not externally validated against real-world longitudinal cohort data.
Ans. Thank you for this important comment. We have now included both internal and external validation results of the proposed dynamic HPV infection–precancer–cancer model in the revised manuscript (Section 4: Materials and Methods, and Section 2: Results). For clarity, the methods and findings are summarized as follows:
4.4 Model Validation
Internal validation was conducted using a simulated Taiwanese cohort with input parameters derived from national datasets and prior studies (Chen et al., 2011; Koong et al., 2006). Key parameters included a cervical cancer incidence of 26.3 per 100,000 woman-years, an HPV prevalence of 12.7%, a 3-year screening interval, and a 52.2% attendance rate.
For external validation, the model was applied to an independent dataset reflecting the Japanese female population, using inputs from IOC/IARC (2023) and Palmar et al. (2024). These included a cervical cancer incidence of 19.9 per 100,000 woman-years, an HPV prevalence of 10.3%, a 2-year screening interval, and an attendance rate of 43.7%. Model-predicted cervical cancer cases were then compared with observed case estimates using a goodness-of-fit (χ²) test.
2.2.1 Internal and External Validation Results
As shown in Supplementary Table S1, the model yielded 2,364 expected cervical cancer cases in the simulated Taiwanese cohort, closely matching the 2,386 cases derived from empirical inputs. The resulting goodness-of-fit test yielded χ² = 0.21 (p = 0.65), indicating an adequate internal model fit.
For external validation using the Japanese female cohort scenario, the model estimated 1,793 cervical cancer cases, compared to the empirical estimate of 1,787 cases, yielding a χ² = 0.02 (p = 0.88). This result supports the generalizability and external validity of our dynamic model across populations with differing demographic and screening profiles.
These validation findings affirm the robustness of our model under both simulated and independent real-world conditions and support its applicability for guiding precision prevention strategies in diverse settings.
Supplementary Table S1. Results of internal and external validation of the dynamic HPV infection-Precancerous-Cancer of Cervix model
|
Internal validation |
External validation |
Information source |
Chen et al., 2011; Koong et al., 2006 |
IOC/IARC 2023; Palmar et al., 2024 |
Incidence of cervical cancer |
26.3 |
19.9 |
Expected number of cervical cancers |
2364 |
1787 |
Prevalence of HPV infection (%) |
12.7 |
10.3 |
Inter-screening interval (year) |
3 |
2 |
Attendance rate (%) |
52.2 |
43.7 |
Number of cervical cancers in simulated cohort |
2386 |
1793 |
χ2 value |
0.21 |
0.02 |
p-value |
0.65 |
0.88 |
(Supplementary Table S1, 4.3.3 Model Validation and 2.2.1 Internal and External Validation)
References
Chen MK, Hung HF, Duffy S, Yen AMF, Chen HH. Cost-effectiveness analysis for Pap smear screening and human papillomavirus DNA testing and vaccination. J Eval Clin Pract. 2011;17(6):1050-1058. doi:10.1111/j.1365-2753.2010.01453.x
ICO/IARC. Papillomavirus, Human. "Related Cancers, Fact Sheet 2023." ICO/IARC information Centre on HPV and cancer (2023).
Koong SL, Yen AMF, Chen THH. Efficacy and cost-effectiveness of nationwide cervical cancer screening in Taiwan. J Med Screen. 2006;13 Suppl 1:S44-47.
Palmer, Matthew R., et al. "The impact of alternate HPV vaccination and cervical screening strategies in Japan: a cost-effectiveness analysis." The Lancet Regional Health–Western Pacific 44 (2024).
Q2. The choice and weighting of biomarkers rely heavily on previously published effect sizes but lack sensitivity analyses to address uncertainty.
Ans. Agreed. We appreciate this important observation. To address this limitation and clarify the study’s purpose, the following paragraph has been added to the revised manuscript (Section 3.4 Limitations):
“This study has several limitations. The distributions and effect sizes of genetic and epigenetic biomarkers were primarily derived from published literature and embedded into a virtual cohort reflective of the cervical cancer risk profile of Taiwanese women. We assumed independence among biomarkers and adopted a uniform joint distribution framework. For each simulated individual, the effects of genetic and epigenetic risk factors were incorporated into the model via a proportional-hazards structure, using additive log-linear transformations. While this approach demonstrates the feasibility of implementing a personalized, risk-guided cervical cancer prevention strategy, it remains dependent on literature-based effect size estimates.
Although prior studies support the additive contribution of polygenes in cancer risk modeling (Pharoah et al., 2008), recent findings suggest that long-range genetic interactions may violate the additivity assumption (Cáceres-Durán et al., 2020). Thus, future integration of population-scale biobank data with our dynamic disease model could help reduce dependence on published estimates and enable exploration of nonlinear and interactive effects among biomarkers through machine learning–based methods. Such integration represents a promising direction for expanding the precision prevention paradigm.”
In addition, to partially address uncertainty in key inputs, we conducted a sensitivity analysis by varying the prevalence of HPV infection—one of the primary risk factors—from 12.7% to 5.1%. We evaluated the robustness of combined prevention strategies under this lower prevalence assumption. The results, presented in Supplementary Table S3, demonstrate that while the overall trend in the effectiveness of prevention strategies remained consistent, the absolute impact was attenuated in the low-prevalence scenario—particularly for women with a risk score percentile below 40%. Notably, the protective effect of HPV vaccination in this subgroup was substantially diminished and did not reach statistical significance.
These findings reinforce the need to continuously refine the model using real-world data and underscore the importance of embedding empirical uncertainty into future risk predictions. They also highlight the potential utility of integrating biobank resources rich in molecular and clinical information to support adaptive, evidence-based cervical cancer prevention strategies.
Supplementary Table S3. Simulated relative risk of cervical cancer by precision prevention strategies with 5.1% prevalence of HPV infection.
|
Relative Risk (95% CI) |
||||||||||||||||||
Risk Score Percentile |
<20% |
20-40% |
40-60% |
60-80% |
>80% |
||||||||||||||
Prevention Strategy |
HPV testing |
HPV testing |
HPV testing |
HPV testing |
HPV testing |
||||||||||||||
- |
+ |
- |
+ |
- |
+ |
- |
+ |
- |
+ |
||||||||||
Pap Smear Screening by Inter-screening Interval |
|||||||||||||||||||
1 yr |
0.75 |
0.53 |
0.58 |
0.49 |
0.58 |
0.49 |
0.51 |
0.48 |
0.47 |
0.45 |
|||||||||
(0.26,2.16) |
(0.39,0.73) |
(0.46,0.73) |
(0.31,0.78) |
(0.46,0.74) |
(0.42,0.57) |
(0.46,0.58) |
(0.41, 0.57) |
(0.41, 0.54) |
(0.41, 0.50) |
||||||||||
3 yr |
0.88 |
0.49 |
0.55 |
0.53 |
0.67 |
0.56 |
0.57 |
0.50 |
0.51 |
0.48 |
|||||||||
(0.32, 2.41) |
(0.36, 0.68) |
(0.44, 0.70) |
(0.34, 0.83) |
(0.53, 0.85) |
(0.48, 0.65) |
(0.49, 0.67) |
(0.45, 0.57) |
(0.44, 0.58) |
(0.43, 0.53) |
||||||||||
5 yr |
1.00 |
0.63 |
0.62 |
0.57 |
0.71 |
0.63 |
0.59 |
0.54 |
0.59 |
0.52 |
|||||||||
|
(0.38,2.66) |
(0.47,0.84) |
(0.40, .95) |
(0.45,0.72) |
(0.57,0.89) |
(0.54,0.73) |
(0.53,0.67) |
(0.46,0.64) |
(0.51,0.67) |
(0.47,0.57) |
|||||||||
HPV Vaccination + Pap Smear Screening by Inter-screening Interval |
|||||||||||||||||||
1 yr |
0.50 |
0.33 |
0.31 |
0.31 |
0.25 |
0.27 |
0.35 |
0.30 |
0.37 |
0.35 |
|||||||||
|
(0.15, 1.66) |
(0.23, 0.46) |
(0.23, 0.42) |
(0.18, 0.53) |
(0.20, 0.31) |
(0.20, 0.37) |
(0.30, 0.40) |
(0.24,0.36) |
(0.33, 0.41) |
(0.30, 0.41) |
|||||||||
3 yr |
0.50 |
0.38 |
0.53 |
0.42 |
0.47 |
0.43 |
0.46 |
0.44 |
0.56 |
0.55 |
|||||||||
|
(0.15,1.66) |
(0.23,0.46) |
(0.42,0.68) |
(0.26,0.68) |
(0.40,0.55) |
(0.33,0.56) |
(0.41,0.52) |
(0.37,0.52) |
(0.51,0.62) |
(0.48,0.63) |
|||||||||
5 yr |
0.88 |
0.63 |
0.57 |
0.49 |
0.63 |
0.53 |
0.59 |
0.50 |
0.69 |
0.52 |
|||||||||
|
(0.32,2.41) |
(0.28,0.54) |
(0.45,0.72) |
(0.31,0.78) |
(0.54,0.73) |
(0.41,0.68) |
(0.53,0.67) |
(0.43,0.59) |
(0.61,0.78) |
(0.32,0.41) |
|||||||||
HPV Vaccination |
|||||||||||||||||||
|
0.97 |
1.00 |
1.04 |
1.02 |
0.94 |
1.04 |
0.97 |
0.96 |
0.94 |
0.94 |
|||||||||
|
(0.75,1.25) |
(0.38,2.66) |
(0.85,1.27) |
(0.70,1.48) |
(0.82,1.07) |
(0.85,1.28) |
(0.87,1.07) |
(0.84,1.09) |
(0.87,1.03) |
(0.84,1.05) |
|||||||||
These results have been included in the revised manuscript (3.4 Limitation)
References
Cáceres-Durán, Miguel Ángel, Ândrea Ribeiro-dos-Santos, and Amanda Ferreira Vidal. "Roles and mechanisms of the long noncoding RNAs in cervical cancer." International journal of molecular sciences 21.24 (2020): 9742.
Hsieh, Hsin‐Ju, Tony Hsiu‐Hsi Chen, and Shu‐Hui Chang. "Assessing chronic disease progression using non‐homogeneous exponential regression Markov models: an illustration using a selective breast cancer screening in Taiwan." Statistics in medicine 21.22 (2002): 3369-3382.
Hsu, Chen-Yang, et al. "Sampling-based Markov regression model for multistate disease progression: applications to population-based cancer screening program." Statistical Methods in Medical Research 29.8 (2020): 2198-2216.
Pharoah, Paul DP, et al. "Polygenes, risk prediction, and targeted prevention of breast cancer." New England Journal of Medicine 358.26 (2008): 2796-2803.
Wu YY, Yen MF, Yu CP, Chen HH. Individually tailored screening of breast cancer with genes, tumour phenotypes, clinical attributes, and conventional risk factors. Br J Cancer. 2013;108(11):2241-2249. doi:10.1038/bjc.2013.202
Wu YY, et al. "Risk assessment of multistate progression of breast tumor with state‐dependent genetic and environmental covariates." Risk analysis 34.2 (2014): 367-379.
Q3. Clarity & Structure: The manuscript is mostly well-organized, with clear progression from background to results and discussion. Tables and figures (e.g., Figure 1, Table 1/2/3) effectively illustrate the key results.
Ans. Thank you for your appreciation.
Q4. The abstract and some text suffer from awkward phrasing (e.g., “disease volution” → “disease evolution”).
Ans. The revised manuscript has been carefully reviewed to correct grammatical errors and improve awkward phrasing throughout the abstract and main text, including changing “disease volution” to “disease evolution”.
Q5. Several grammatical errors and redundancies hinder readability (e.g., “remarkable higher probability” should be “markedly higher”; "Nature history" → "Natural history").
Ans. The revised manuscript has been carefully edited to correct grammatical errors and remove redundancies, including changing “remarkable higher probability” to “markedly higher probability” and correcting “Nature history” to “Natural history.” These amendments have been made to enhance overall readability.
Q6. Section transitions (e.g., between Results 2.1 and 2.2) could be smoother to improve logical flow.
Ans. Agreed. Thank you for the valuable suggestion. To enhance the logical flow and ensure a smoother transition between Sections 2.1 and 2.2, the following bridging paragraph has been added to the revised manuscript:
“Building on the molecular data presented above—including the population distribution and state-specific effect sizes of each genetic and epigenetic biomarker—we constructed a simulated cohort of 300,000 women, embedding these molecular characteristics within the natural disease course of cervical neoplasia. This virtual cohort reflects the progression from normal epithelium to LSIL, HSIL, and ultimately invasive cervical carcinoma. To evaluate the influence of HPV infection status and molecular risk factors on cervical lesion progression, we integrated data abstracted from the literature into this representative cohort of Taiwanese women. Utilizing the natural history parameters reported by Chen et al. (2011), we projected the disease burden in the absence of preventive interventions, thereby establishing a baseline scenario for subsequent evaluation of precision prevention strategies.”
This revision has been incorporated in the manuscript to improve the clarity and continuity between the results subsections.
Q7. Methodological Rigor: The multistate model design, including backward transitions, is methodologically strong. Incorporation of relative risk and population frequency in the risk score formulation is robust.
Ans. Thank you for your recognition.
Q8. No mention of model calibration or internal validation.
Ans. The methods and results of internal and external validation have been addressed in the revised manuscript in response to Q1.
Q9. The assumption of uniform distribution for SNPs and methylation patterns, based on literature estimates, may not reflect population heterogeneity accurately.
Ans. Agreed, we have included this argument as limitation of current study and suggest a better use of rich molecular information collected in biobank to capture the potential correlation inherited in multiple genetic and epigenetic biomarkers and non-linear relationship between the using machine learning approaches. This has been included in the revised manuscript in response to Q2.
Q10. The equation for risk score (Formula 1) lacks clarity in explaining λ₀ and β parameters; also, it doesn't justify why additive log-linear modeling was chosen.
Ans. The process for the derivation of risk scores considering the genetic and epigenetic molecular markers were further elaborated by using the six illustrative cases with the detailed profiles provided.
“To illustrate this dynamic transition from HPV infection to precancerous lesions and finally to cervical cancer, six representative cases were selected (Figure 3, Cases A to F). The first three cases (Cases A, B, and C; Figure 3a) represent women with HPV infection, while the remaining three (Cases D, E, and F; Figure 3b) are HPV-negative. The complete genetic and DNA methylation profiles of these six individuals are summarized in the profile table (Table 3).”
“The ith individual’s risk score given HPV status (j=0 HPV negative, j=1 HPV positive) was calculated using the formula:
RiskScoreij = log(λ₀j) + ∑(βᵢ × xᵢ)
where λ₀j represents baseline hazard rate given the jth HPV status, xᵢ represents the genetic or epigenetic profile of a given woman, βᵢ denotes the log-transformed effect size reported in prior studies (see Table 1 and Table 2). Risk scores were then transformed into percentiles within the simulated cohort, and both the raw scores and percentiles are presented in Table 3.
For example, Case A’s risk score is computed as:
(−2.00) + (−0.15) +(4.21)+ (0.75) = 2.81,
based on the following molecular features: HPV positivity, Pre-mir-218 rs1113452 (AA), C13ORF18 positivity and DAPK positivity.”
(2.2.3 Scenarios of Personalized Risk Assessment and 4.3.1 Genetic Biomarkers Associated with the Evolution of Cervical Cancer)
Table 3. Detailed profiles and derived risk scores and percentile of risk scores and the effectiveness of precision prevention guided by the risk percentile compared with universal triennial Pap smear of the six illustrative cases
Profiles |
RR / Baseline Hazard Rate of HPV (λ₀j) |
Regression Coefficients (β)/ ln(λ₀j) |
Case A |
Case B |
Case C |
Case D |
Case E |
Case F |
CD28+17 (TT) INFG+874 (AA) |
0.78 |
-0.25 |
|
|
|
|
V |
V |
Pre-mir-218 rs11134527 (GG) |
0.73 |
-0.31 |
|
V |
|
|
|
V |
Pre-mir-218 rs11134527 (AA) |
0.86 |
-0.15 |
V |
|
|
V |
|
|
LAMB3 rs2566 (TT) |
1.80 |
0.59 |
|
|
|
V |
|
|
LAMB3 rs2566 (CT) |
1.59 |
0.46 |
|
V |
V |
|
V |
|
CASP8 -652 6N del/del |
0.53 |
-0.63 |
|
|
|
|
|
|
CASP8 -652 6N del/ins |
0.75 |
-0.29 |
|
|
|
|
|
|
DUT rs3784621 (CC) |
1.54 |
0.43 |
|
|
|
V |
|
|
DUT rs3784621 (CT) |
1.33 |
0.29 |
|
|
|
|
V |
|
GTF2H4 rs2894054 (AA) |
0.11 |
-2.21 |
|
|
|
|
|
|
GTF2H4 rs2894054 (AG) |
0.57 |
-0.56 |
|
|
|
|
|
|
OAS3 rs12302655 (AA) |
1.57 |
0.45 |
|
|
|
|
|
|
SULF1 rs4737999 (AA) |
0.59 |
-0.53 |
|
|
|
|
|
|
SULF1 rs4737999(AG) |
0.59 |
-0.53 |
|
|
|
|
|
|
IFNG rs11177074 (CC) |
1.35 |
0.30 |
|
|
|
|
|
|
IFNG rs11177074 (CT) |
1.78 |
|
|
|
|
|
|
|
POLN rs17132382 (CT) |
2.16 |
0.77 |
|
|
|
|
|
|
POLN rs17132382 (TT) |
2.47 |
0.90 |
|
|
|
V |
|
|
TMC8 rs9893818 (AA) |
1.57 |
0.45 |
|
|
|
|
|
|
CCNA1 |
42.08 |
3.74 |
|
|
|
|
|
|
C13ORF18 |
67.66 |
4.21 |
V |
|
|
|
|
|
SFRP |
3.92 |
1.37 |
|
|
|
|
V |
|
DAPK |
2.11 |
0.75 |
V |
V |
|
V |
V |
|
HIC-1 |
2.72 |
1.00 |
|
V |
|
V |
|
|
HIN-1 |
2.13 |
0.76 |
|
|
|
|
|
|
MGMT |
1.29 |
0.25 |
|
|
|
|
|
|
RAR-beta |
3.58 |
1.28 |
|
V |
|
|
|
|
RASSF1A |
2.77 |
1.02 |
|
|
|
|
|
|
SHP-1 |
6.38 |
1.85 |
|
|
|
|
|
|
Twist |
1.80 |
0.59 |
|
|
|
|
|
|
Persistent HPV infection status |
λ₀j Positive (j=1): 0.135 Negative (j=0): 0.051 |
ln(λ₀j) Positive (j=1): -2.00 Negative (j=0): -2.98 |
Positive |
Positive |
Positive |
Negative |
Negative |
Negative |
Risk Score (log(λ0j)+ ) |
- |
- |
2.81 |
1.16 |
-1.55 |
0.57 |
-0.36 |
-3.53 |
Percentile of Risk |
- |
- |
>80% |
>80% |
40-60% |
60-80% |
60-80 |
<20 |
RR (vs Triennial Pap Smear) |
- |
- |
0.61 (0.53-0.71)1 p<0.001 |
0.61 (0.53-0.71)1 p<0.001 |
0.83 (0.71-0.97)2 P=0.02 |
0.55 (0.48, 0.64)3 p < 0.001 |
0.55 (0.48, 0.64)3 p < 0.001 |
1.02 (0.71, 1.46)4 p = 0.93 |
1 HPV vaccination + HPV test + Annual Pap smear
2 HPV test + Triennial Pap smear
3 HPV vaccination + Triennial Pap smear
4 Pap smear at 6-year interval
Leveraging the log-linear model has been addressed in the answer to Q2 and has been included as a limitation in the revised manuscript (3.4 Limitation)
Q11. Expand the method section to include Model validation/calibration procedures.
Ans. The methods of model validation have been included in the revised manuscript as addressed in Q1.
Q12. Sensitivity analyses biomarker prevalence and RR inputs.
Ans. This has been addressed and included in the revised manuscript, as noted in the response to Q2 earlier.
Q13. Discuss the ethical implications and feasibility of genetic testing in national screening programs.
Ans.
While genetic and epigenetic biomarkers hold significant promise for enabling personalized cervical cancer prevention, incorporating genetic testing into national screening programs presents important ethical and practical challenges. These challenges are best examined through the lens of the four core bioethical principles: autonomy, beneficence, non-maleficence, and justice.
- Respect for autonomy requires that individuals are fully informed and able to freely choose whether to undergo genetic testing. This entails rigorous informed consent procedures and transparent communication about the implications, limitations, and potential psychological impacts of receiving genetic risk information. Opt-in or opt-out mechanisms must be designed to protect individual choice and avoid coercion.
- Beneficence obliges the screening program to demonstrably improve health outcomes. Genetic testing should provide actionable insights—such as identification of high-risk individuals who would benefit from intensified surveillance or early intervention—that lead to meaningful clinical benefits.
- Non-maleficence emphasizes the importance of minimizing harm. Potential risks include false positives, overdiagnosis, stigmatization, anxiety, and unnecessary interventions. To mitigate these harms, the program must ensure that test results are accompanied by high-quality genetic counseling, and that thresholds for clinical action are clearly evidence-based.
- Justice demands fair access and equitable distribution of benefits and burdens. Screening infrastructure must be designed to avoid exacerbating existing disparities based on socioeconomic status, geography, or digital literacy. Policymakers must also consider the affordability of follow-up care and the long-term psychosocial consequences for individuals labeled as genetically high-risk.
Although technically feasible, the implementation of genetic testing within a population-wide program requires careful policy design, public engagement, and ethical safeguards. It must be coupled with health system readiness, workforce training in genetic counseling, data privacy protections, and ongoing evaluation of clinical utility and social acceptability (Spencer et al., 2022).
These ethical considerations and feasibility issues are addressed in the revised manuscript under Section 3.3 (Cervical Cancer Precision Prevention Guided by Risk Score Percentile).
Reference
Spencer, Scott J., and Stephanie M. Fullerton. "Population genomic screening: Ethical considerations to guide age at implementation." Frontiers in Genetics 13 (2022): 899648.
Q14. Compare existing models to contextualize innovation and impact.
Ans.
Multiple approaches have been proposed to advance the goal of precision and personalized cervical cancer prevention, particularly through the integration of artificial intelligence and predictive modeling.
For instance, Wang et al. developed the AI-based Cervical Cancer Screening System (AICCS), which applied deep learning to enhance diagnostic accuracy in cytology grading, demonstrating improved sensitivity over conventional Pap smear interpretation. He et al. employed machine learning to predict spontaneous regression of LSIL, highlighting non-traditional risk factors such as sleep quality, thus expanding the scope of individualized risk assessment. Rothberg et al. proposed a demographic-based risk prediction model that utilized routine clinical variables to personalize cervical screening intervals by predicting CIN2+ risk. Elvatun et al. conducted a comparative evaluation of existing cervical cancer risk prediction models across diverse populations, revealing heterogeneity in model performance but identifying several algorithms with promising cross-population generalizability. Additionally, Dong et al. introduced the SMART-HPV model, which integrated high-risk HPV (hrHPV) genotyping to stratify risk with high accuracy, particularly in settings with limited screening access.
While these models represent significant progress, most focus on binary classification tasks—predicting the presence or absence of specific endpoints such as CIN2+ or cervical cancer—limiting their applicability in simulating lesion evolution and in guiding early-stage preventive interventions.
By contrast, the current study introduces a dynamic, multistate natural history model of cervical neoplasia progression, spanning from normal epithelium through LSIL and HSIL to invasive carcinoma. Our model uniquely incorporates genetic and epigenetic biomarker profiles, enabling risk stratification beyond demographic or HPV genotype alone. This structure supports the simulation of personalized prevention strategies—including HPV vaccination, testing, and Pap smear screening—tailored to individual molecular risk percentiles.
The key innovation lies in the model’s ability to simulate temporal disease progression under various intervention scenarios, allowing for evaluation of both early prevention and screening optimization strategies. This approach addresses a critical gap in existing models by offering a more granular and biologically informed framework for precision cervical cancer prevention, thus providing actionable insights for both clinical decision-making and public health planning.
This comparison and positioning are discussed in Section 4.4 (Comparison with Existing Models and Innovation) of the revised manuscript
Reference
Dong, Binhua, et al. "Development, validation, and clinical application of a machine learning model for risk stratification and management of cervical cancer screening based on full-genotyping hrHPV test (SMART-HPV): a modelling study." The Lancet Regional Health–Western Pacific 55 (2025).
He, Simin, et al. "Predictive models for personalized precision medical intervention in spontaneous regression stages of cervical precancerous lesions." Journal of Translational Medicine 22.1 (2024): 686.
Elvatun, Severin, et al. "Cross-population evaluation of cervical cancer risk prediction algorithms." International Journal of Medical Informatics 181 (2024): 105297.
Wang, Jue, et al. "Artificial intelligence enables precision diagnosis of cervical cytology grades and cervical cancer." Nature Communications 15.1 (2024): 4369.
Rothberg, Michael B., et al. "A risk prediction model to allow personalized screening for cervical cancer." Cancer Causes & Control 29 (2018): 297-304.
Q15. Consider including cost-effectiveness projections for the precision prevention strategies proposed.
Ans. While evaluating the cost-effectiveness of personalized precision screening is paramount, it is beyond the scope of the current study. Chen et al. (2011) conducted a cost-effectiveness analysis of cervical cancer prevention strategies in Taiwan using a probabilistic Markov decision model. Their study concluded that annual Pap smear screening remains the most cost-effective approach, while combining HPV DNA testing with triennial screening is economically favorable under reasonable willingness-to-pay thresholds. Additionally, HPV vaccination coupled with triennial screening could become cost-effective if vaccine prices significantly decrease. Personalized screening tailors both the screening method and frequency to an individual's specific risk profile. By integrating genetic and epigenetic biomarkers with conventional cytology and HPV testing, women at higher risk can receive more intensive surveillance. Such a risk-stratified strategy may maximize clinical benefits, enhance cost-effectiveness, reduce unnecessary interventions, and promote true precision in preventive care. This represents a crucial area for future research.
This has been included in the revised manuscript. (3.3 Cervical Cancer Precision Prevention Guided by Risk Score Percentile)
Reference
Chen MK, et al.: Cost-effectiveness analysis for Pap smear screening and human papillomavirus DNA testing and vaccination. J Eval Clin Pract. 2011;17(6):1050-1058.
Q16. Recommendations for Improvement: Clarify language throughout the manuscript and ensure professional editing.
Ans. The revised manuscript has been thoroughly reviewed by professor of this field to correct grammatical errors and improve readability.

Round 2
Reviewer 1 Report
Comments and Suggestions for Authors
The authors have adequately answered the recommendations and clarifications. The revised version is greatly improved and can be published pending final check by editorial team.
Reviewer 2 Report
Comments and Suggestions for Authors
All concerns have been adequately addressed